# Dynamics and Representation Structure of Local Approximations to Gradient-Based Learning in Linear Recurrent Neural Networks

**Ezekiel Williams** [1][2][3]   **Alexandre Payeur** [1][2]   **Guillaume Lajoie** [1][2]

## Abstract

Biological and neuromorphic recurrent neural networks (RNNs) are subject to spatial and temporal locality constraints on the information that can plausibly be used during learning. A common strategy to satisfy these constraints is to modify gradient descent by neglecting non-local terms to varying degrees, as in random feedback local online (RFLO) learning and truncated backpropagation through time (tBPTT). However, the learning dynamics of these algorithms, and how they compare with BPTT, remain poorly understood. We apply dynamical systems theory to data-aligned linear RNNs—whose dynamics can be separated into orthogonal modes—to compare stationary solutions, stability properties, and convergence rates, finding qualitatively distinct behaviour for RFLO versus BPTT and one-step tBPTT. We further observe that the solutions learned by RFLO are restricted to low-rank perturbations of initial parameters, a result which holds beyond the data-aligned setting. Our work provides analytical insight into how locality constraints shape learning dynamics, with implications for neuroscientific models of learning and alternative optimization approaches for RNNs.

## 1. Introduction

A deeper understanding of learning in recurrent neural networks (RNNs) is important for both neuroscience and machine learning (ML). RNNs are a standard model of brain circuits (Barak, 2017). They provide a flexible framework for sequence modelling, relevant to many applied domains (De Mulder et al., 2015; Sahoo et al., 2019; Durstewitz et al.,

2023), and have recently seen a resurgence in the AI community (Tran et al., 2018; Katharopoulos et al., 2020; Peng et al., 2023). In both contexts, these networks are routinely trained using gradient-based optimization (Goodfellow et al., 2016; Richards & Kording, 2023). However, computing exact gradients in RNNs is inherently non-local (Zenke & Neftci, 2021; Bredenberg et al., 2023): parameter updates may depend on spatially and temporally distant information, as in backpropagation through time (BPTT) (Werbos, 1990), or be spatially non-local but temporally local at an extra memory cost, as in real-time recurrent learning (RTRL) (Williams & Zipser, 1989). Such non-locality complicates training of neuromorphic systems (Zenke & Neftci, 2021) and is widely viewed as biologically implausible (Lillicrap et al., 2020; Lillicrap & Santoro, 2019).

These drawbacks have motivated the search for alternative learning rules that approximate gradient descent using only local information (Roth et al., 2018; Murray, 2019; Bellec et al., 2020; Ellenberger et al., 2025). A common approach is to neglect non-local terms in the true gradient, yielding the random feedback local online (RFLO) algorithm (Murray, 2019) and e-prop (Bellec et al., 2020) when applied to RTRL, and the one-step version of truncated BPTT (tBPTT) (Williams & Peng, 1990) when applied to BPTT. However, these local learning rules generally do not correspond to the gradient of any objective function and therefore need not follow descent dynamics on the original loss. As a result, theoretical understanding of how closely their learning dynamics resemble those of exact gradient descent—including their fixed points, stability, and convergence properties—remains limited, despite recent progress (Liu et al., 2025; Murray, 2019).

To address this, we build on prior mathematical analyses of learning in neural networks. Some insights have been obtained in nonlinear settings, for example about the impact of biased gradients on generalization in RNNs (Liu et al., 2022) or the effectiveness of large nonlinear RNNs at memorizing data (Allen-Zhu et al., 2019). Closer to our interest in local learning rules, Bordelon & Pehlevan (2023) used statistical-physics methods to study biologically plausible rules in infinite-width, nonlinear feedforward networks. However, most theoretical analyses of learning dynamics

[1]Department of Mathematics and Statistics, Université de Montréal, Canada [2]Mila Québec, Canada [3]Imperial College London, United Kingdom. Correspondence to: Ezekiel Williams <e.williams1@imperial.ac.uk>.

*Proceedings of the 43rd International Conference on Machine Learning*, Seoul, South Korea. PMLR 306, 2026. Copyright 2026 by the author(s).

have relied on linear networks for tractability, either in feed-forward (Saxe et al., 2014; Braun et al., 2022; Dominé et al., 2023; 2025) or recurrent (Hardt et al., 2018; Li et al., 2022; Schuessler et al., 2020; Proca et al., 2025) architectures. Despite linear computations, gradient descent in linear networks gives rise to highly nonlinear learning dynamics (Saxe et al., 2014), making them a useful setting for theoretical study. Moreover, linear RNNs are of independent interest: in neuroscience, they have been widely used to model motor systems (Hennequin et al., 2014; Lara et al., 2018; Logiaco et al., 2021), while in ML, linear recurrent state-space models (SSMs) have recently emerged as memory-efficient alternatives to transformers (Gu et al., 2022). We therefore restrict our analysis to linear RNNs.

Our approach uses a student-teacher setting in the data-aligned regime (Proca et al., 2025), in which neural dynamics decompose into orthogonal, non-interacting, modes. We derive ordinary differential equations (ODEs) describing the infinite-data, continuous-time limit of RFLO, tBPTT and BPTT. We compare their fixed points, stability, and local spectral properties to characterize their asymptotic learning speeds. Our main findings are:

1. For tBPTT and BPTT, the fixed points organize into two one-dimensional manifolds: an optimal manifold that is stable and a non-optimal manifold that is unstable (saddle).

2. In contrast, RFLO admits only an optimal manifold of fixed points, with stability properties that differ qualitatively from BPTT and tBPTT due to its random feedback structure.

3. Numerical experiments show that the theory can extend beyond the data-aligned regime, after an initial transient period, provided the teacher's dynamics consist of non-interacting exponential decays.

4. RFLO restricts learned solutions in linear RNNs to be low-rank. This result holds without the data-alignment assumption, and numerical experiments further suggest that this low-rank restriction may also affect other local algorithms, like tBPTT and, to a lesser extent, e-prop.

## 2. Methods

This section presents the mathematical derivations underlying our results. We first introduce the student-teacher RNN model and the three learning algorithms together with their parameter update rules. Next, we describe the data-alignment assumption and show how it can be used to diagonalize the tBPTT and RFLO updates, extending Proca et al. (2025)'s approach for BPTT. Finally, we take the small-learning rate limit to yield the ODEs that form the basis of our analysis.

### 2.1. Student-Teacher Model

We consider a student-teacher setup in which both networks are linear, discrete-time RNNs driven by the same Gaussian white noise process. The student is defined by

$$h_{t+1} = Wh_t + Bx_t,$$
$$y_{t+1} = Ah_{t+1} \quad (1)$$

where $x_t \in \mathbb{R}^m$ is drawn i.i.d. from $\mathcal{N}(0, \mathbf{I})$ for $t = 0, 1, \ldots, T-1$, $h \in \mathbb{R}^n$ is the hidden state and $y \in \mathbb{R}^o$ is the output. The Gaussian white noise assumption is standard in theoretical analyses of neural networks, and typically justified by invoking upstream processes that whiten and normalize the inputs (Saxe et al., 2014; 2019; Proca et al., 2025). The teacher is defined by the same equations, but with different parameters and possibly a different hidden-layer size. Teacher quantities are denoted by a $\star$ (e.g., $h^\star \in \mathbb{R}^{n_\star}$). We assume $h_0 = 0$ and $h_0^\star = 0$ throughout.

The learning objective is to minimize the expected $\mathcal{L}^2$ distance between the student and teacher outputs at the final time step $T$:

$$L_T = \frac{1}{2}||y_T - y_T^\star||^2,$$
$$\theta^* = \operatorname*{argmin}_{\theta} \mathbb{E}[L_T], \quad (2)$$

where $\theta = \{A, W, B\}$ represents the parameters of the student network and the expectation $\mathbb{E}$ is taken over the input distribution. In Appendix H, we show that our main results generalize to the sequence loss $\mathcal{L} = \frac{1}{2T}\sum_{t=1}^{T}||y_t - y_t^\star||^2$.

### 2.2. Learning Algorithms

We study the learning dynamics resulting from different algorithms applied to the objective in Eq. 2, namely BPTT (Werbos, 1990), tBPTT (Williams & Peng, 1990), and RFLO (Murray, 2019). For each of these algorithms, the learning rule follows the general form $\theta_{k+1} = \theta_k - \eta \Delta\theta_k$, where $\Delta\theta_k$ denotes the parameter update at iteration $k$. Assuming interchangeability of gradient and expectation, i.e. $\nabla_\theta \mathbb{E}[L_T] \equiv \mathbb{E}[\nabla_\theta L_T]$, BPTT yields the following update for the recurrent weights of the student model (Appendix B):

$$\Delta W = \sum_{t=1}^{T}(W^{T-t})^\top A^\top \mathbb{E}[\varepsilon_T h_{t-1}^\top] \quad (3)$$

where $\varepsilon_T = y_T - y_T^\star$ is the output error. In tBPTT, error propagation is truncated to $\tau$ steps:

$$\Delta_\tau W = \sum_{t=T-\tau+1}^{T}(W^{T-t})^\top A^\top \mathbb{E}[\varepsilon_T h_{t-1}^\top]. \quad (4)$$

RFLO (Murray, 2019) approximates RTRL (Williams & Zipser, 1989) by replacing the product of Jacobians in the

true gradient with a fixed, diagonal matrix and using random feedback to propagate errors (Lillicrap et al., 2016) (Appendix B):

$$\Delta_{\mathrm{RFLO}}W = \sum_{t=1}^{T}(\widehat{W}^{T-t})^{\top} R^{\top} \mathbb{E}[\varepsilon_T h_{t-1}^{\top}], \qquad (5)$$

where $R$ is a fixed random feedback matrix with the same dimensions as $A$, and $\widehat{W}$ can adopt different values depending on the specific modification made to RTRL (see below). The updates for the input weights $B$ are obtained by replacing $h_{t-1}$ with $x_{t-1}$ in Eqs. 3-5. The update for $A$ is identical across all algorithms. This is shown in Appendix B, along with additional derivational details on the three learning rules.

The learning rules in Eqs. 3-5 can be expanded by computing the cross-correlations $\mathbb{E}[\varepsilon_T h_{t-1}^{\top}]$ (and $\mathbb{E}[\varepsilon_T x_{t-1}^{\top}]$ for $B$). Defining

$$\mathcal{E}_t = AW^t B - A_\star W_\star^t B_\star, \qquad (6)$$

the recurrent weight updates can be written as (see Appendix C for the remaining parameters):

$$\Delta W = \sum_{t=1}^{T-1}\sum_{s=0}^{t-1}[AW^s]^{\top}\mathcal{E}_t[W^{t-s-1}B]^{\top}, \qquad (7)$$

$$\Delta_\tau W = \sum_{t=1}^{T-1}\sum_{s=0}^{\min\{\tau-1,t-1\}}[AW^s]^{\top}\mathcal{E}_t[W^{t-s-1}B]^{\top}, \qquad (8)$$

$$\Delta_{\mathrm{RFLO}} W = \sum_{t=1}^{T-1}\sum_{s=0}^{t-1}[R\widehat{W}^s]^{\top}\mathcal{E}_t[W^{t-s-1}B]^{\top}. \qquad (9)$$

**Remark:** Several variants of RFLO arise depending on the choice of $\widehat{W}$ (Appendix B). In the original formulation (Murray, 2019), $\widehat{W} \propto \mathbf{I}$, with a proportionality constant equal to one minus the time constant of the network dynamics. In our setting, the dynamics are instantaneous (i.e., the time constant is one), so we instead treat the scaling as a free parameter and set

$$\widehat{W} = \hat{w}\mathbf{I}. \qquad (10)$$

We refer to this variant as RFLO in what follows. Alternative choices include a fixed diagonal matrix or, to solely delete non-local Jacobian terms, a version where $\widehat{W}$ is given by the diagonal part of $W$. This latter version corresponds to e-prop (Bellec et al., 2020) applied to rate networks, rather than spiking RNNs as in its original formulation, and we thus refer to it as e-prop in the rest of the paper.

### 2.3. Data-Aligned Linear RNNs

Learning dynamics in linear neural networks are non-trivial, and assumptions are typically required to render their anal-ysis tractable. The data-alignment assumption is one such approach, and utilizes a change of basis to jointly diagonalize the input data (or teacher) and network model. For an $n$ dimensional network, this reduces learning dynamics to the study of $n$ decoupled modes each with dynamics restricted to its own three-dimensional space.

The data-alignment assumption begins with the input-output correlation matrix of the data (Saxe et al., 2014; 2019; Proca et al., 2025). In our setting, where the output is evaluated only at the final time step $T$, this correlation is $\Sigma_t^\star = \mathbb{E}[y_T^\star x_t^\top]$. Following Proca et al. 2025, we assume that $\Sigma_t^\star$ admits the decomposition $\Sigma_t^\star = US_tV^\top$, where $U$ and $V$ are orthogonal matrices, and $S_t$ is diagonal for all $t$. For simplicity, we assume equal input and output dimensions; the extension to unequal dimensions is straightforward. This resembles a singular value decomposition, except that the diagonal elements of $S_t$ are not constrained to be non-negative. Then each left singular vector $u_i$ couples only to the corresponding right singular vector $v_i$ at every time step, with no interaction between distinct mode pairs $(u_i, v_j)$ for $i \neq j$.

This structure imposes constraints on the teacher. From the linear dynamics, we have $\Sigma_t^\star = A_\star W_\star^{T-1-t} B_\star$ (Proca et al., 2025). If $W_\star$ admits an orthogonal decomposition

$$W_\star = P_\star \bar{W}_\star P_\star^\top, \qquad (11)$$

with $\bar{W}_\star$ diagonal, and if

$$A_\star = U\bar{A}_\star P_\star^\top \qquad (12)$$

$$B_\star = P_\star \bar{B}_\star V^\top, \qquad (13)$$

with $\bar{A}_\star$ and $\bar{B}_\star$ diagonal, then $\Sigma_t^\star = U\bar{A}_\star \bar{W}_\star^{T-1-t}\bar{B}_\star V^\top$, which satisfies the assumed decomposition. In this case, we say that the teacher is *mode-aligned*: its recurrent dynamics couple corresponding input–output modes without mixing across modes. For simplicity, we assume that the recurrent dimension matches the input and output dimensions.

Given a mode-aligned teacher, we then say that the data-alignment assumption holds—i.e. the student-teacher model is data-aligned—if, at initialization, the student is also mode-aligned and each student mode lines up with a unique teacher input-output mode:

$$A_0 = U\bar{A}_0 P^\top \qquad (14)$$

$$B_0 = P\bar{B}_0 V^\top \qquad (15)$$

$$W_0 = P\bar{W}_0 P^\top, \qquad (16)$$

where $\bar{A}_0$, $\bar{B}_0$ and $\bar{W}_0$ are diagonal, and $P$ is an orthogonal matrix diagonalizing $W_0$. More generally, the sign of any column of $U$ or $V$ can be flipped without affecting the decomposition. Thus, the alignment between singular vectors can be either parallel or antiparallel. As we show in the

next section, this alignment leads to learning dynamics that decouple across modes.

## 2.4. Diagonalizing tBPTT and RFLO

Using the data-alignment assumptions, we show that the learning dynamics can be diagonalized not just for BPTT (Proca et al., 2025), but also for tBPTT and RFLO. This is achieved by expressing the update rules (Eqs. 7-9) in the aligned basis and exploiting the orthogonality of the transformation matrices. For RFLO, this requires specifying the feedback matrix $R$ in a manner consistent with the aligned structure ($\widehat{W} = \hat{w}\mathbf{I}$ is already aligned). To permit full diagonalization, we take $R = U\bar{R}P^\top$, where $\bar{R}$ is a diagonal matrix with random elements. Under these assumptions, the RFLO updates take the form (see Appendix D for BPTT and tBPTT)

$$\Delta_{\mathrm{RFLO}}\bar{W} = \sum_{t=1}^{T-1}\sum_{s=0}^{t-1} \hat{w}^s \bar{R}\overline{\mathcal{E}}_t \bar{B}\bar{W}^{t-s-1},$$

$$\Delta_{\mathrm{RFLO}}\bar{A} = \sum_{t=0}^{T-1} \overline{\mathcal{E}}_t \bar{B}\bar{W}^t, \qquad (17)$$

$$\Delta_{\mathrm{RFLO}}\bar{B} = \sum_{t=0}^{T-1} \hat{w}^t \bar{R}\overline{\mathcal{E}}_t,$$

where

$$\overline{\mathcal{E}}_t = \bar{A}\bar{W}^t\bar{B} - \bar{A}_\star \bar{W}_\star^{\,t}\bar{B}_\star \qquad (18)$$

is the diagonalized version of Eq. 6. All overlined quantities are diagonal. Because the RHS of each equation in Eq. 17 is diagonal, the learning updates become diagonal so, if the student network is initialized according to Eqs. 14-16, each mode will evolve independently during learning.

## 2.5. Deriving ODEs

To derive the ODEs used in our results we rely on two limits. First, similar to Zucchet & Orvieto (2024), we consider the limit of long input stimuli, $T \to \infty$. In our case this yields a closed form for the learning dynamics. For each decoupled mode, we denote the student parameters by the triplet $(a, b, w)$—the $i^{th}$ diagonal entries of $\bar{A}, \bar{B}, \bar{W}$ for $i \in \{1, \ldots, n\}$—and the corresponding teacher parameters by $(a_\star, b_\star, w_\star)$. For notational simplicity, we drop the $i$ subscript. Using standard properties of infinite geometric series, the learning update for $a$—which is identical across learning rules—becomes

$$\Delta a \to \frac{ab^2}{1-w^2} - \frac{a_\star bb_\star}{1-ww_\star}. \qquad (19)$$

The RFLO updates for $b$ and $w$ become (Appendix E):

$$\Delta_{\mathrm{RFLO}}w \to \frac{\hat{a}ab^2w}{(1-\hat{w}w)(1-w^2)} - \frac{\hat{a}a_\star bb_\star w_\star}{(1-\hat{w}w_\star)(1-w_\star w)}$$

$$\Delta_{\mathrm{RFLO}}b \to \frac{\hat{a}ab}{1-\hat{w}w} - \frac{\hat{a}a_\star b_\star}{1-\hat{w}w_\star}, \qquad (20)$$

where $\hat{a}$ denotes the diagonal element of $\bar{R}$ corresponding to $a$. For BPTT, we obtain (Appendix E)

$$\Delta w \to \frac{a^2b^2w}{(1-w^2)^2} - \frac{aa_\star bb_\star w_\star}{(1-ww_\star)^2}$$

$$\Delta b \to \frac{a^2b}{1-w^2} - \frac{aa_\star b_\star}{1-ww_\star}. \qquad (21)$$

For one-step tBPTT, we obtain (Appendix E):

$$\Delta_\tau w \to \frac{a^2b^2w}{1-w^2} - \frac{aa_\star bb_\star w_\star}{1-ww_\star} \qquad (22)$$

$$\Delta_\tau b \to a^2b - aa_\star b_\star.$$

Finally, to simplify the analysis we take the small-learning-rate limit $\eta \to 0$, in which the discrete updates are well approximated by ODEs: $\dot{\theta} = -\Delta\theta$ (Ali et al., 2019).

## 2.6. Comparing Theory and Experiments

To evaluate the predictive value of the theory and investigate its limitations, we trained students RNNs that were not data-aligned to learn mode-aligned teacher networks. Teachers satisfied $m_\star = n_\star = o_\star$, whereas students could have larger recurrent layers for greater generality. For the teacher, we chose $A_\star = B_\star = \mathbf{I}$ and $W_\star$ diagonal, such that $U = V = P_\star = \mathbf{I}$. We used distinct positive eigenvalues for the teacher recurrent matrix, to facilitate student-teacher comparisons. Student RNNs were initialized from a small-variance Gaussian distribution, following (Saxe et al., 2019).

To estimate the student mode values $(a, b, w)$ at each training epoch, we computed

$$D_A = U^\top A\hat{P}, \; D_B = \hat{P}^\top BV, \; D_W = \hat{P}^{-1}W\hat{P}, \quad (23)$$

based on the data-alignment assumptions. For RFLO, we similarly computed $D_R = U^\top R\hat{P}$. Here, $D_A, D_B, D_W$, and $D_R$ estimate $\bar{A}, \bar{B}, \bar{W}$ and $\bar{R}$, respectively, and $\hat{P}$ diagonalizes $W$ at the given epoch. Because $W$ was not constrained to be symmetric in the simulations, $\hat{P}$ is generally non-orthogonal. For convenience, we sorted the $n_\star$ dominant student modes—those with the largest recurrent eigenvalues—in decreasing order, and matched them to the correspondingly sorted teacher modes. The quantities $(D_{Aii}, D_{Bii}, D_{Wii})$ then provide numerical estimates of the student mode $(a, b, w)$ associated with the $i^{\text{th}}$ teacher mode, while $D_{Rii}$ estimates $\hat{a}$.

We quantified alignment separately for the recurrent, input/output, and random feedback matrices. If the recurrent alignment assumptions are satisfied, then $\hat{P}^\top \hat{P} = \tilde{\mathbf{I}}$, where $\tilde{\mathbf{I}}$ is block-diagonal with an $n_\star \times n_\star$ identity block in the top-left corner and an arbitrary lower-right block when $n > n_\star$. We define recurrent alignment as

$$\mathcal{A}_{\text{rec}} = 1 - \frac{||(\hat{P}^\top \hat{P} - \tilde{\mathbf{I}})_{:n_\star}||_F}{||(\hat{P}^\top \hat{P})_{:n_\star}||_F}, \tag{24}$$

where subscript $:n_\star$ denotes the first $n_\star$ rows of the matrix. If input, output, or random feedback alignment assumptions hold, and if $o = m$ (as in our experiments), then $D_A \odot D_B^\top$ and $D_R$ are rectangular diagonal matrices. We define their respective alignment as

$$\mathcal{A}_{\text{in/out}} = 1 - \frac{||D_A \odot D_B^\top - \text{Diag}(D_A \odot D_B^\top)||_F}{||D_A \odot D_B^\top||_F}, \tag{25}$$

$$\mathcal{A}_{\text{random}} = 1 - \frac{||D_R - \text{Diag}(D_R)||_F}{||D_R||_F}. \tag{26}$$

All measures lie in $[0, 1]$, where 1 denotes perfect alignment.

To generate theory predictions, we selected a training epoch, initialized the appropriate ODE system (Section 2.5) using the estimated student mode values at that epoch, and numerically integrated it. The deviation between theory and experiment (theory error) was defined as the time-averaged 1-norm difference between theoretical and numerical trajectories, and depends on the choice of initialization epoch (see Section 3.5).

## 3. Results

### 3.1. Fixed-Point Manifolds

We first determine the location of the fixed points for BPTT, tBPTT with $\tau = 1$ and RFLO. For this, we set the expected updates in the limit $T \to \infty$ (Eqs. 19-22) to zero, and solve for $w$, $a$ and $b$, assuming $w_\star, a_\star, b_\star, \hat{w}, \hat{a} \neq 0$. In this limit, the denominators in Eqs. 19-22 are guaranteed not to vanish (Appendix E), so the solutions are well-defined.

The fixed points organize into two manifolds (Fig. 1). First, all three algorithms admit fixed points that form the lines defined by $ab = a_\star b_\star$ and $w = w_\star$, which minimize the loss; we refer to this one-dimensional manifold as optimal. This manifold has two branches. Second, BPTT and tBPTT (but not RFLO) admit an additional manifold of fixed points given by the line $a = b = 0$. Along this manifold, the expected loss satisfies $\mathbb{E}[L_T] \to \sum^n a_\star^2 b_\star^2 (1 - w_\star^2)^{-1}/2$, where the sum runs over the $n$ modes. As this loss is strictly positive, we refer to this manifold as non-optimal. For RFLO, the condition $\Delta_{\text{RFLO}}b = 0$ enforces $b \neq 0$ due to the random feedback, which rules out this manifold. Thus,

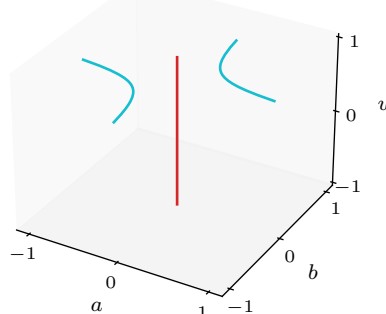

*Figure 1.* Fixed-point curves for the three learning algorithms. Optima ($ab = a_\star b_\star$, $w = w_\star$) are in cyan and non-optima ($a = b = 0$, $w$ free) in red. BPTT and tBPTT share the same fixed-point structures, whereas RFLO lacks the non-optimal line. Parameters: $w_\star = 0.7$, $a_\star = 0.4$, $b_\star = 0.25$, $\hat{a} = 0.2$, $\hat{w} = 0.3$.

unlike BPTT and tBPTT, RFLO admits only optimal fixed points.

### 3.2. Stability

For each algorithm, each eigenmode obeys a nonlinear three-dimensional ODE system, where $-(\Delta a, \Delta b, \Delta w)$ defines the vector field. For BPTT, this vector field is a gradient flow, whereas for tBPTT and RFLO the resulting ODEs are not, in general, gradients of any function. The vector field of tBPTT is similar to that of BPTT in our setup (Fig. 2). By contrast, RFLO's vector field differs markedly from the others and, in particular, lacks the non-optimal manifold, which in BPTT and tBPTT shapes the flow near $a = b = 0$. BPTT's and tBPTT's vector fields suggest that the line at $a = b = 0$ is a saddle, while the other fixed points are stable. The stability properties of RFLO's manifolds are less obvious—consider for instance the vector field near the lower-left branch in Fig. 2 (right)—and thus require a more detailed analysis.

To analyze stability in more detail, we linearize the ODE dynamics around the fixed points. The Jacobians are evaluated along the optimal and non-optimal manifold for BPTT and one-step tBPTT, and along the optimal manifold for RFLO. The optimal manifold can be parametrized as $(a, b, w) = (s, a_\star b_\star s^{-1}, w_\star)$ with $s \in \mathbb{R} \setminus \{0\}$. The eigenvalues of the Jacobian characterize stability along the manifolds. Since these are lines of fixed points, at least one eigenvalue is zero. The two other eigenvalues may depend on the learning rule and the location on the manifold (see Appendix F for detailed computations).

For one-step tBPTT, on the non-optimal manifold, the two nonzero eigenvalues are $\pm a_\star b_\star (1 - ww_\star)^{-1/2}$. Because

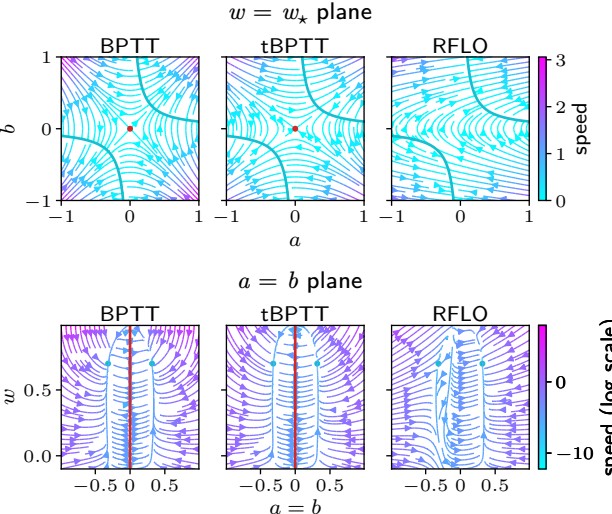

*Figure 2.* Vector fields for all learning rules (left: BPTT; center: tBPTT ($\tau = 1$); right: RFLO). Top row: slice through 3D space at $w = w_\star$. Bottom row: slice at $a = b$. Same parameters as in Fig. 1. Cyan lines and dots: optimal manifold; red lines and dots: non-optimal manifold.

the limit $T \to \infty$ requires $|ww_\star| < 1$ (Appendix E), these eigenvalues are real positive and negative. Therefore, this manifold is a saddle, like with BPTT (Appendix F). On the optimal manifold, the characteristic polynomial of the Jacobian $\mathcal{J}_1$ is

$$|\mathbf{I}\lambda - \mathcal{J}_1| = \lambda^3 + \lambda^2(s^2 + a_\star^2 b_\star^2 s^{-2} m_\star + a_\star^2 b_\star^2 m_\star^2)$$
$$+ \lambda(s^2 a_\star^2 b_\star^2 m_\star^2 + a_\star^4 b_\star^4 s^{-2} m_\star^2), \quad (27)$$

where we defined $m_\star = (1 - w_\star^2)^{-1}$. The nonzero eigenvalues are the roots of the quadratic polynomial with linear coefficient $x = s^2 + a_\star^2 b_\star^2 s^{-2} m_\star + a_\star^2 b_\star^2 m_\star^2$ and constant term $y = s^2 a_\star^2 b_\star^2 m_\star^2 + a_\star^4 b_\star^4 s^{-2} m_\star^2$, i.e., $\lambda_\pm = \frac{1}{2}(-x \pm \sqrt{x^2 - 4y})$. Because $x, y > 0$ for any choice of parameters and $s$, we see that the optimal manifold is stable. Moreover, our numerical explorations suggest that the nonzero eigenvalues are real (Fig. 3).

For RFLO, we only need to evaluate the Jacobian (Appendix F, Eq. 114) on the optimal manifold. Still using $x$ as a shorthand for the linear coefficient and $y$ for the constant factor, we have

$$x = a_\star^2 b_\star^2 s^{-2} m_\star$$
$$+ \hat{a}s\hat{m}_\star + \hat{a}a_\star^2 b_\star^2 s^{-1}(1 - \hat{w}w_\star^3)\hat{m}_\star^2 m_\star^2 \quad (28)$$
$$y = (\hat{a}^2 a_\star^2 b_\star^2 + \hat{a}a_\star^4 b_\star^4 s^{-3}) m_\star^2 \hat{m}_\star^2, \quad (29)$$

where $\hat{m}_\star = (1 - \hat{w}w_\star)^{-1}$. Note that $\hat{m}_\star > 0$ and $(1 - \hat{w}w_\star^3) > 0$ for any $w_\star$ and $\hat{w}$ because $|\hat{w}w_\star| < 1$ (Appendix E) and because $|w_\star| < 1$ when the teacher is stable. Here, $x$ and $y$ are not necessarily positive, because the products $\hat{a}s$, $\hat{a}s^{-1}$ can be negative. We thus have two cases to analyze:

**Case $\hat{a}s > 0$:** in this case, $x > 0$ and $y > 0$, and the point $s$ on the optimal manifold is stable whenever it has the same sign as $\hat{a}$. Because of this, for RFLO there is always one optimal branch that is stable regardless of the sign of $\hat{a}$ (Fig. 3). Our numerical explorations suggest that the eigenvalues are real.

**Case $\hat{a}s < 0$:** in this case, both $x$ and $y$ could be negative. Since $x$ appears with a minus sign in the quadratic formula, the largest eigenvalue $\lambda_+$ can become unstable (Fig. 3). Moreover, when the product $a_\star^2 b_\star^2$ becomes large enough, a portion of the optimal manifold can be associated with unstable oscillations (Fig. 3B), a behavior absent in BPTT and one-step tBPTT.

Additionally, for both $\hat{a}s < 0$ and $\hat{a}s > 0$, increasing $\hat{w}$ primarily pushes the real parts of eigenvalues away from the origins $(s, \text{Re}(\lambda_+)) = (0, 0)$ and $(s, \text{Re}(\lambda_-)) = (0, 0)$ (Supp. Fig. 7). Thus, in contrast to BPTT and one-step tBPTT, for which the optimal manifold is stable and the non-optimal manifold is a saddle, RFLO exhibits sign-dependent stability along the optimal manifold and can display unstable or oscillatory regimes.

### 3.3. Convergence Rates

The eigenvalue analysis further allows us to compare the asymptotic convergence rates of the different algorithms—i.e., how quickly learning converges in a neighbourhood of a fixed point. When the nonzero eigenvalues have negative real parts, typically associated with zero imaginary part (Fig. 3), the eigenvalue with the largest real part ($\lambda_+$) determines the slowest rate of convergence (Fig. 3, top row). Among the algorithms, BPTT exhibits the fastest convergence rate, whereas RFLO is the slowest over most of the optimal manifold. Only for $s$ values very close to $0$ does RFLO achieve the fastest convergence rate. Modes initialized near fixed points with small $a^*$ and large $b^*$ can therefore be learned quickly with RFLO. For BPTT, the slowest convergence rate occurs at $a^* = b^* = \pm\sqrt{|a_\star b_\star|}$, indicated by the vertical dashed lines in Fig. 3. Finally, we note that when $|w_\star|$ is small, tBPTT's convergence rates are mostly similar to those of BPTT (Fig. 3C).

### 3.4. Parameter Space Trajectories

The above analysis is local in parameter space $(a, b, w)$. We now ask how these features manifest in the global learning dynamics of the three updates rules. To address this question, we numerically integrated the dynamics given by Eqs. 19-22. For illustration, we initialized the dynamics close to one branch of the optimal manifold. BPTT and its truncated variant converge quickly to this manifold (Fig. 4). In contrast, RFLO fails to converge to the nearest branch and, after a long excursion in parameter space, converges to

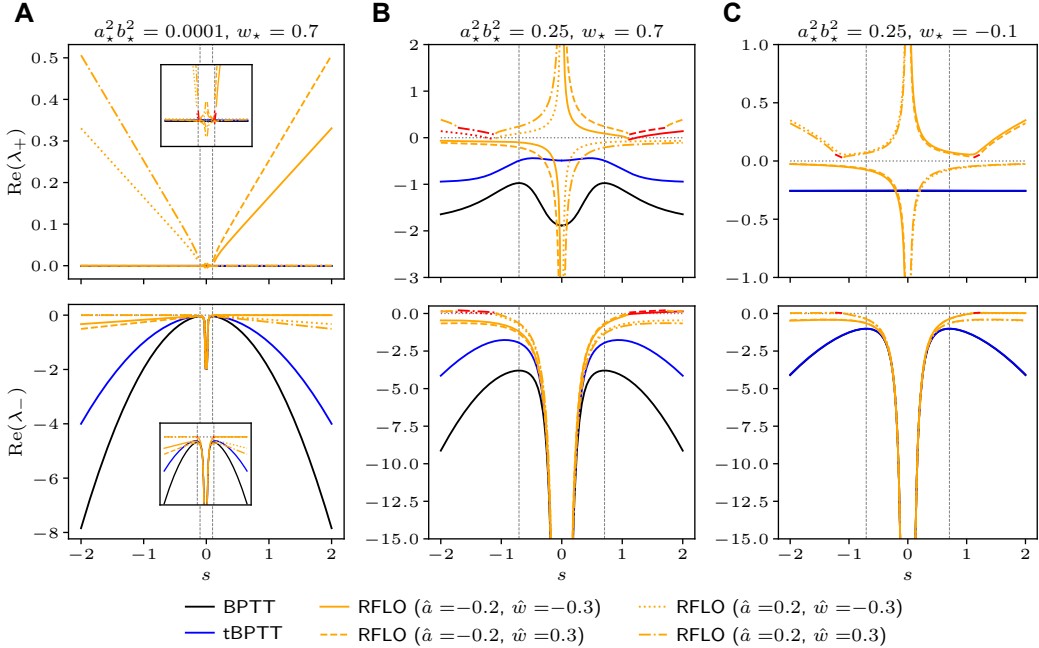

*Figure 3.* Real part of the largest ($\lambda_+$, top) and smallest ($\lambda_-$, bottom) eigenvalues on the optimal manifold $ab = a_\star b_\star, w = w_\star$. (A) $a_\star$, $b_\star$ and $w_\star$ are the same as in Fig. 1. Note that the RFLO curves close to $\text{Re}(\lambda_-) = 0$ are actually slightly positive at $s = 2$. Insets: magnified view near $s = 0$. (B) Same as A, but with $a_\star^2 b_\star^2$ increased to 0.25. (C) Same as B but with $w_\star$ decreased to $-0.1$. Values for $\hat{a}$ and $\hat{w}$ are $\pm$ those used in Fig. 1. Red curve sections for RFLO corresponds to eigenvalues with nonzero imaginary part. Dashed vertical lines indicate $s = \pm\sqrt{|a_\star b_\star|}$, the maxima of BPTT's real eigenvalues. Because negative exponents of $s$ appear in the eigenvalues, some ranges of the y-axis were cropped to improve readability.

the opposite branch, leading to a drastic increase in the learning time. Thus, the distinct fixed-point structure of RFLO, together with its location-dependent stability along the optimal manifold, has a substantial impact on the resulting learning trajectories.

### 3.5. Predictions Beyond the Data-Aligned Regime

We compare theoretical predictions for RFLO when a student RNN that is not data-aligned learns a mode-aligned teacher (Appendix I.1 for experimental details). As described in the Methods (Section 2.6), the theory ODEs can be initialized at any point during training using the estimated RNN student modes. We find that the theory matches the experimental data reasonably well, particularly for the two largest modes, when initialized after a brief transient period at the start of training (Fig. 5A). During training, the student progressively aligns with the teacher, with the recurrent and input/output matrices becoming almost fully aligned; the random feedback matrix, however, remains partially misaligned (Fig. 5B). A representative training run shows the stronger agreement for the two dominant modes (modes 1 and 2) compared to the smaller ones (Fig. 5C). There is also a qualitative tendency for larger modes to be learned more quickly, although we do not quantify this effect. Better agreement between theory and experiment later

in training, and increasing alignment during training, also hold for BPTT and tBPTT (Appendix J, Figs. 8-9).

### 3.6. Representation Structure of Solutions

In this section, we explore the solutions that local approximations to gradient descent can learn from the perspective of matrix rank. We relax the data-alignment assumption from earlier sections. For RFLO, we show that whenever $\widehat{W} = \hat{w}\mathbf{I}$—a setting that includes the original formulation of RFLO (Murray, 2019) as a special case (Appendix B)—learning is restricted to low-rank perturbations of the initial weights for $W$ and $B$. This result is summarized in the following proposition.

**Proposition 3.1.** *Let $W_k$ and $B_k$ be the recurrent and input matrices, respectively, of a linear RNN at the $k^{th}$ iteration of RFLO. Then*

$$W_K = W_0 + \sum_{i=1}^{o} r_i q_i^\top, \tag{30}$$

$$B_K = B_0 + \sum_{i=1}^{o} r_i q_i^{(b)\top}, \tag{31}$$

*where $r_i$ is the $i^{th}$ row of $R$, $q_i$ is the $i^{th}$ row of $-\sum_{k,t,s} \eta \hat{w}^s \mathcal{E}_t^k B_k^\top W_k^{t-s-1\top}$, and $q_i^{(b)}$ the $i^{th}$ row of*

**A**

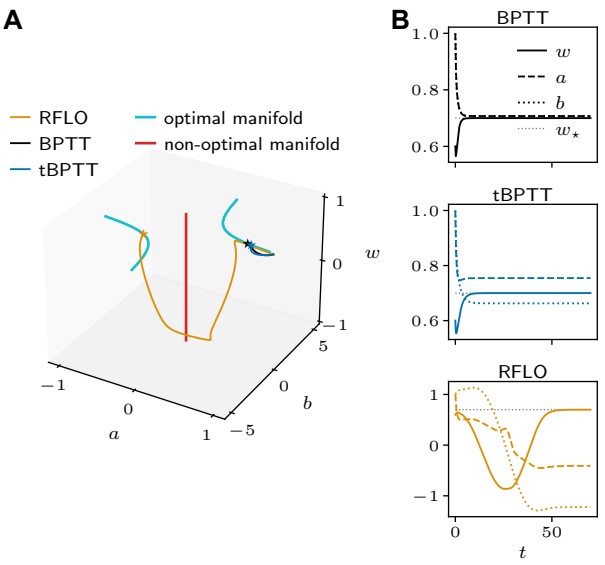

**B**

*Figure 4.* Parameter space trajectories. (A) Trajectories. All algorithms start at the same point $(a, b, w) = (1, 1, 0.6)$. The termination points are indicated by star symbols. (B) Parameters as a function of time for the trajectories in A. Parameters: $w_\star = 0.7$, $a_\star^2 b_\star^2 = 0.25$, $\hat{w} = 0.3$, $\hat{a} = -0.2$.

$-\sum_{k,t,s} \eta \hat{w}^s \mathcal{E}_t^k$. *Lastly, o is the output dimension.*

This means the rank of the weight matrix changes learned by RFLO are bounded by the output dimension $o$.

We next asked whether other local learning algorithms are similarly restricted in rank. To address this question, we trained linear student networks with tBPTT, RFLO and e-prop to learn a teacher with a range of timescales (see Appendix I.2 for details). We then compared the normalized eigenspectra of the training-induced change in $W$ to those obtained with BPTT (Fig. 6). For the task considered, the learned eigenvalues were effectively all real. All local algorithms learned lower rank solutions than BPTT. As predicted by our theory, RFLO learned rank-1 perturbations. tBPTT performed similarly, while e-prop produced solutions of intermediate rank between RFLO/tBPTT and BPTT. We attribute RFLO's relatively poor performance to two inherent features. First, it relies on a scaled identity matrix for recurrent error propagation, unlike e-prop, which uses a diagonal matrix. Second, it feeds back output errors through a random matrix, whereas tBPTT and BPTT use $A^\top$. The higher rank tended to correlate with higher task performance (see Supp. Fig.10). Interestingly, RFLO and e-prop were less stable than tBPTT and BPTT: out of the 20 simulations performed for each algorithm, 13 converged for RFLO and 12 for e-prop. Overall, these results suggest that, at least in linear RNNs, rank restrictions imposed by local learning rules limit the expressivity of learned solutions while reducing performance compared to exact gradient descent.

## 4. Discussion

We find that RFLO learning dynamics differ markedly from those of tBPTT and BPTT in our model, while still converging to optimal solutions. RFLO has fewer stable points on the optimal manifold, which can lead to large detours in parameter space. Moreover, RFLO is not only globally inefficient but also asymptotically inefficient, typically exhibiting slower convergence rates than the other two algorithms. In nonlinear networks, these phenomena could increase the likelihood of becoming trapped in local minima or even diverging.

Despite these limitations, our finding that RFLO-trained linear RNNs converge to solutions that are low-rank perturbations of the initial connectivity may provide a mechanistic basis for the low-rank connectivity structures often invoked in neuroscience (Mastrogiuseppe & Ostojic, 2018; Beiran et al., 2021). Moreover, the fact that the rank is bounded by the output dimension may help explain why tasks involving more timescales than output dimensions can be difficult to learn (Murray, 2019). RFLO variants such as e-prop can converge to higher-rank solutions, but whether this follows from the heterogeneity or the plasticity of the eligibility trace time constants (Eq. 51) remains unclear.

Most sequence-modelling tasks require temporal credit assignment, i.e., determining how past parameter values affect present task errors. Our finding that one-step tBPTT and BPTT behave similarly shows that data-aligned problems do not require temporal credit assignment, because one-step tBPTT cannot assign credit across multiple time steps by construction. Previous work has linked long timescales to difficult credit assignment (Lillicrap & Santoro, 2019). In contrast, the similarity predicted under the data-alignment hypothesis, which can still generate long autocorrelation timescales, suggests that the interaction between dynamic modes also plays an important role. By extension, we posit that any theory of temporal credit assignment cannot rely on data-alignment assumptions.

The main limitation of our theory is the data-alignment assumption. One might expect the theory to fail at predicting mode dynamics during training even when only the student network is misaligned. Surprisingly, however, there is often a regime in which alignment is poor but the theory remains predictive, at least for certain modes. More work is needed to determine the underlying mechanisms, but one possibility is that well-predicted modes align before full matrix-level alignment is achieved. Whether the theory generalizes to more challenging tasks—e.g., when the input is not white noise or the teacher has degenerate eigenvalues—and to cases where the teacher is not mode-aligned, remains to be investigated. We speculate that the theory might still work for teacher modes that are weakly interacting. For a more complete picture, future research on local learning rules

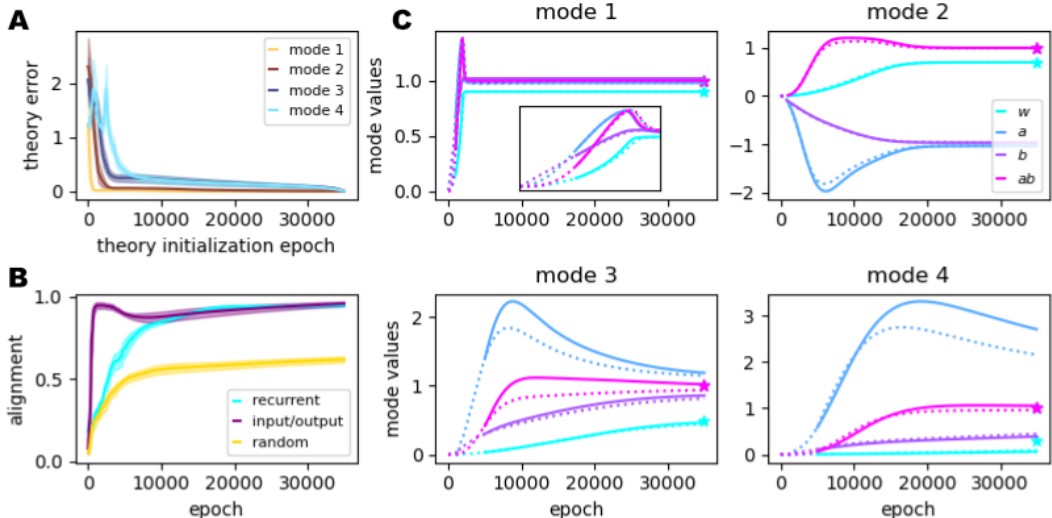

*Figure 5.* Comparison of data-aligned theory with non-data-aligned numerical experiments for RFLO. (A) Theory error for each mode as a function of initialization epoch (see Methods). (B) Alignment as a function of training epoch. In panels A and B, curves show mean $\pm$ standard error (shaded region) over 6 seeds. (C) Comparison of experimental dynamics (dotted lines) and theory (solid) for the student modes corresponding to each of the four teacher modes, ranked from largest to smallest (mode 1 is largest). A single representative learning trajectory is shown. Inset for mode 1 shows the first 2500 epochs. See Section 2.6 and Appendix I.1 for details, and Appendix J for corresponding BPTT and tBPTT results.

will need to go beyond the data-alignment assumption, perhaps by extending MLP mean-field approaches (Bordelon & Pehlevan, 2023) to RNNs.

## Acknowledgements

E.W. wishes to thank the other members of the Lajoie lab, Nicolas Zucchet, and João Sacramento for helpful discussions. GL acknowledges support from the Canada-CIFAR AI Chair Program and from the NSERC Discovery program.

## Impact Statement

This paper presents work whose goal is to advance the fields of Machine Learning (ML) and Theoretical Neuroscience by studying abstract mathematical models. Therefore it should not have direct negative impacts. However, as with any research on "dual-use" technology like machine learning, concern for indirect negative impacts is warranted. The authors note, in particular, that neuromorphic advances due to a deeper understanding of local learning rules could result in ML being employed on small devices with potential military applications. In light of the increasingly prominent role of AI in condemnable acts of violence, surveillance, and oppression, the authors wish to mention this potential downstream impact and highlight the need for increased education and action to contend with military-related, and other, irresponsible research applications.

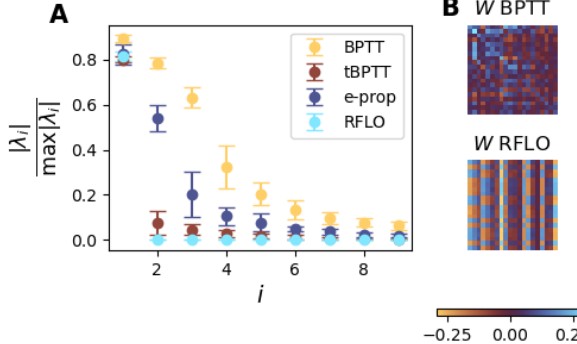

*Figure 6.* (A) Spectrum of the change in $W$ relative to initialization (i.e., spectrum of $W_K - W_0$, where $K$ is the final training iteration) learned by BPTT, tBPTT, RFLO, or e-prop, on a linear student-teacher task with a single output dimension. $Y$-axis: absolute value of the eigenvalues; $x$-axis: eigenvalue index, ordered from largest to smallest magnitude. All spectra are normalized by the absolute value of the largest eigenvalue. Curves show mean $\pm$ standard deviation over converged training runs: 13 for RFLO; 12 for e-prop; 20 for the other algorithms. (B) Example recurrent matrix $W$ after training with BPTT (top) and RFLO (bottom).

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

## A. Notation

1. $\mathbf{I}$: identity matrix.

2. Trace: Tr

3. Transpose of matrix $A$: $A^\top$

4. $L^2$ norm: $||\cdot||$

5. Frobenius norm: $||\cdot||_F$

## B. The Three Learning Rules

The main text considers three learning rules: backpropagation through time (BPTT), truncated BPTT and random feedback local online learning (RFLO). We detail them here for a fixed input realization and our terminal loss $L_T$.

**BPTT**  For our linear RNN, BPTT yields the updates

$$\Delta W := \nabla_W L_T = \sum_{t=1}^{T} \delta_t h_{t-1}^\top,$$

$$\Delta B := \nabla_B L_T = \sum_{t=1}^{T} \delta_t x_{t-1}^\top, \tag{32}$$

$$\Delta A := \nabla_A L_T = \epsilon_T h_T^\top,$$

where the credit-assignment vector $\delta_t$ satisfies the backward recursion

$$\delta_t = W^\top \delta_{t+1}$$
$$\delta_T = A^\top \epsilon_T \tag{33}$$

for $t = T-1, \ldots, 1$. In the linear case, this recursion admits the closed-form solution

$$\delta_t = (W^{T-t})^\top A^\top \epsilon_T. \tag{34}$$

**tBPTT**  Truncated BPTT restricts error backpropagation to the last $\tau - 1$ steps only, according to our definition of $\tau$. The tBPTT credit-assignment vector is

$$\delta_t^{(\tau)} = \begin{cases} \delta_t, & T - \tau + 1 \leq t \leq T, \\ 0, & \text{otherwise.} \end{cases} \tag{35}$$

Substituting into the BPTT gradients yields

$$\Delta_\tau W = \sum_{t=T-\tau+1}^{T} \delta_t h_{t-1}^\top, \tag{36}$$

$$\Delta_\tau B = \sum_{t=T-\tau+1}^{T} \delta_t x_{t-1}^\top, \tag{37}$$

$$\Delta_\tau A = \Delta A. \tag{38}$$

Only the summation limits differ from BPTT.

**RFLO**  RFLO is derived as an approximation to real-time recurrent learning (RTRL). In RTRL, the gradients are

$$\frac{dL_T}{dW_{ij}} = \sum_k [A^\top \varepsilon_T]_k P_{ij,T}^k, \tag{39}$$

$$\frac{dL_T}{dB_{ij}} = \sum_k [A^\top \varepsilon_T]_k Q_{ij,T}^k, \tag{40}$$

where the tensors $P_{ij,t}^k$ and $Q_{ij,t}^k$ evolve according to

$$P_{ij,T}^l = \delta_{li} h_{T-1,j} + \sum_k W_{lk} P_{ij,T-1}^k, \tag{41}$$

$$Q_{ij,T}^l = \delta_{li} x_{T-1,j} + \sum_k W_{lk} Q_{ij,T-1}^k, \tag{42}$$

for a linear RNN with instantaneous dynamics, like ours. The original RFLO neglects the sums on the right-hand side and replaces $A$ by a fixed random matrix (feedback alignment) (Lillicrap et al., 2016). In our setting, however, neglecting these terms eliminates the temporal integration of past neural states. Instead, we impose the diagonal approximation $W_{lk} = \hat{w}\delta_{lk}$, which removes nonlocal interactions while preserving temporal traces. The recursions then reduce to

$$P_{ij,t}^k = \delta_{ki} h_{t-1,j} + \hat{w} P_{ij,t-1}^k, \tag{43}$$

$$Q_{ij,t}^k = \delta_{ki} x_{t-1,j} + \hat{w} Q_{ij,t-1}^k, \tag{44}$$

with initial conditions $P_{ij,0}^k = 0$ and $Q_{ij,0}^k = 0$. Solving these recursions yields

$$P_{ij,T}^k = \delta_{ki} \sum_{t=1}^T \hat{w}^{T-t} h_{t-1,j}, \tag{45}$$

$$Q_{ij,T}^k = \delta_{ki} \sum_{t=1}^T \hat{w}^{T-t} x_{t-1,j}. \tag{46}$$

Replacing $A^\top$ by a fixed random matrix $R^\top$, the updates become

$$\Delta_{\mathrm{RFLO}} W = \sum_{t=1}^T \hat{w}^{T-t} R^\top \varepsilon_T h_{t-1}^\top, \tag{47}$$

$$\Delta_{\mathrm{RFLO}} B = \sum_{t=1}^T \hat{w}^{T-t} R^\top \varepsilon_T x_{t-1}^\top, \tag{48}$$

$$\Delta_{\mathrm{RFLO}} A = \Delta A, \tag{49}$$

which corresponds to the credit-assignment vector

$$\delta_t^{(\mathrm{RFLO})} = \hat{w}^{T-t} R^\top \varepsilon_T. \tag{50}$$

When $\hat{w} = 0$, the updates reduce to the original RFLO rule for our model; thus, $\hat{w}$ plays a role analogous to the time constant in (Murray, 2019).

*Variants.* One may instead take $W_{lk} = \hat{w}_l \delta_{lk}$ (random diagonal) or $\widehat{W}_{lk} = W_{ll}\delta_{lk}$ (linear RNN version of e-prop). In the latter case,

$$P_{ij,T}^k = \delta_{ki} \sum_{t=1}^T W_{kk}^{T-t} h_{t-1,j}, \tag{51}$$

and

$$\frac{dL_T}{dW_{ij}} \approx [A^\top \varepsilon_T]_i \sum_{t=1}^T W_{ii}^{T-t} h_{t-1,j}. \tag{52}$$

The rule remains local, with $W_{ii}$ interpretable as a learnable neuron-specific time constant.

## C. Expected Updates

The updates computed in Appendix B were derived for a single input realization. We now compute their expected value under an i.i.d. standard Gaussian input process. We assume that differentiation and expectation can be interchanged, i.e., $\nabla_\Theta \mathbb{E}[L_T] = \mathbb{E}[\nabla_\Theta L_T]$. For BPTT, it is more convenient to compute $\nabla_\Theta \mathbb{E}[L_T]$, whereas for tBPTT and RFLO we directly evaluate $\mathbb{E}[\Delta_\tau \Theta]$ and $\mathbb{E}[\Delta_{\mathrm{RFLO}} \Theta]$.

**BPTT** We first compute the expected loss. Since

$$y_T = \sum_{t=0}^{T-1} AW^t B x_{T-1-t} \qquad y_T^\star = \sum_{t=0}^{T-1} A_\star W_\star^t B_\star x_{T-1-t}, \tag{53}$$

we obtain

$$\mathbb{E}[L_T] = \frac{1}{2} \mathbb{E}\left[\mathrm{Tr}\{(y_T - y_T^\star)(y_T - y_T^\star)^\top\}\right] = \frac{1}{2} \mathrm{Tr}\left\{ \sum_{t,t'=0}^{T-1} \mathcal{E}_t \mathbb{E}[x_{T-1-t} x_{T-1-t'}^\top] \mathcal{E}_{t'}^\top \right\}, \tag{54}$$

where

$$\mathcal{E}_t = AW^t B - A_\star W_\star^t B_\star. \tag{55}$$

Since the input is an i.i.d. standard Gaussian, $\mathbb{E}[x_{T-1-t} x_{T-1-t'}^\top] = \delta_{tt'} I$, which yields

$$\mathbb{E}[L_T] = \frac{1}{2} \mathrm{Tr}\left\{ \sum_{t=0}^{T-1} \mathcal{E}_t \mathcal{E}_t^\top \right\} = \frac{1}{2} \sum_{\tau=0}^{T-1} \|\mathcal{E}_t\|_F^2. \tag{56}$$

To compute the gradients, we note that the differential of a scalar function of a matrix variable $f(X)$ is $df = \mathrm{Tr}\{\nabla_X f^\top dX\}$. We have

$$d\mathbb{E}[L_T] = \sum_{t=0}^{T-1} \mathrm{Tr}\{\mathcal{E}_t^\top d\mathcal{E}_t\}. \tag{57}$$

For $W$, with $A$ and $B$ fixed,

$$d\mathcal{E}_t = A d(W^t) B = \sum_{k=0}^{t-1} AW^k dW W^{t-1-k} B, \tag{58}$$

where we used

$$dW^t = \sum_{k=0}^{t-1} W^k dW W^{t-1-k}. \tag{59}$$

Thus,

$$d\mathbb{E}[L_T] = \sum_{t=0}^{T-1} \sum_{k=0}^{t-1} \mathrm{Tr}\{W^{t-1-k} B \mathcal{E}_t^\top AW^k dW\}, \tag{60}$$

which implies

$$\nabla_W \mathbb{E}[L_T] = \sum_{t=1}^{T-1} \sum_{k=0}^{t-1} (AW^k)^\top \mathcal{E}_t (W^{t-1-k} B)^\top. \tag{61}$$

(The sum starts at $t = 1$ since $d\mathcal{E}_0 = 0$.)

For $B$,

$$d\mathcal{E}_t = AW^t dB \Rightarrow \nabla_B \mathbb{E}[L_T] = \sum_{t=0}^{T-1} (AW^t)^\top \mathcal{E}_t. \tag{62}$$

For $A$,

$$d\mathcal{E}_t = dAW^t B \Rightarrow \nabla_A \mathbb{E}[L_T] = \sum_{t=0}^{T-1} \mathcal{E}_t (W^t B)^\top. \tag{63}$$

**tBPTT** We compute the expectation of Eqs. 36 and 37. These involve $\mathbb{E}[\varepsilon_T h_{t-1}^\top]$ and $\mathbb{E}[\varepsilon_T x_{t-1}^\top]$. Using

$$h_t = \sum_{\tau=0}^{t-1} W^\tau B x_{t-1-\tau}, \qquad \epsilon_T = y_T - y_T^\star = \sum_{\tau=0}^{T-1} \mathcal{E}_\tau x_{T-1-\tau} \tag{64}$$

we obtain

$$\mathbb{E}[\epsilon_T h_{t-1}^\top] = \sum_{\tau=0}^{t-2} \mathcal{E}_{T-1-\tau} B^\top (W^{t-2-\tau})^\top \tag{65}$$

and

$$\mathbb{E}[\epsilon_T x_{t-1}^\top] = \mathcal{E}_{T-t}. \tag{66}$$

Using these,

$$\Delta_\tau W = \sum_{t=T-\tau+1}^{T} \sum_{t'=0}^{t-2} (AW^{T-t})^\top \mathcal{E}_{T-1-t'} (W^{t-2-t'} B)^\top. \tag{67}$$

We reindex the sums to express the update in terms of $\mathcal{E}_s$, so as to match the structure of the BPTT update. We first change variable to $s = T - 1 - t'$, yielding

$$\Delta_\tau W = \sum_{t=T-\tau+1}^{T} \sum_{s=T-t+1}^{T-1} (AW^{T-t})^\top \mathcal{E}_s (W^{t-T+s-1} B)^\top. \tag{68}$$

We now interchange the order of summation to get

$$\Delta_\tau W = \sum_{s=1}^{T-1} \sum_{t=\max\{T-\tau+1, T-s+1\}}^{T} (AW^{T-t})^\top \mathcal{E}_s (W^{t-T+s-1} B)^\top. \tag{69}$$

Finally, we change variable to $k = T - t \Rightarrow 0 \le k \le \min\{\tau - 1, s - 1\}$:

$$\Delta_\tau W = \sum_{s=1}^{T-1} \sum_{k=0}^{\min\{\tau-1, s-1\}} (AW^k)^\top \mathcal{E}_s (W^{s-k-1} B)^\top, \tag{70}$$

which is the expression in the main text once the summation variables are renamed.

For $B$, using $s = T - t$,

$$\Delta_\tau B = \sum_{s=0}^{\tau} (W^\top)^s A^\top \mathcal{E}_s. \tag{71}$$

**RFLO** Since the RFLO updates differs from the BPTT updates only by the substitutions $W \to \widehat{W}$ and $A \to R$ in the feedback pathway, the expected updates for $W$ and $B$ follow directly from the BPTT expressions under the same substitutions.

## D. Diagonalization

We diagonalize the expected updates under the data-alignment assumption (Eqs. 11-16 in the main text). Using $W = P\overline{W}P^\top$, we have $W^t = P\overline{W}^t P^\top$, by orthogonality of $P$. A similar expression holds for the teacher. Using these expressions together with Eqs. 11-16, the error matrix $\mathcal{E}_t$ becomes

$$\mathcal{E}_t = U\overline{\mathcal{E}}_t V^\top, \tag{72}$$

with

$$\overline{\mathcal{E}}_t = \bar{A}\bar{W}^t\bar{B} - \bar{A}_\star \bar{W}_\star^{\,t}\bar{B}_\star, \tag{73}$$

which enters all update expressions. The learning dynamics can be written as

$$\dot{W} = -P\overline{\Delta W}P^\top \tag{74}$$

$$\dot{B} = -P\overline{\Delta B}V^\top \tag{75}$$

$$\dot{A} = -U\overline{\Delta A}P^\top, \tag{76}$$

where $\overline{\Delta\theta}$ denotes the diagonal version of update $\Delta\theta$. The orthogonality of $P, V, U$ then yields the fully diagonalized dynamics:

$$\dot{\overline{W}} = -\overline{\Delta W} \tag{77}$$

$$\dot{\overline{B}} = -\overline{\Delta B} \tag{78}$$

$$\dot{\overline{A}} = -\overline{\Delta A}. \tag{79}$$

We now derive the diagonal form of the updates for each algorithm.

**BPTT**  For $W$, we replace Eqs. 11-16 and the transformed error (Eq. 72) in the expected update (Eq. 61) to get

$$\Delta W = \sum_{t=1,s=0}^{T-1,t-1} [U\bar{A}\bar{W}^s P^\top]^\top U(\bar{A}\bar{W}^t\bar{B} - \bar{A}_\star\bar{W}_\star^{\,t}\bar{B}_\star)V^\top[P\bar{W}^{t-s-1}\bar{B}V^\top]^\top. \tag{80}$$

Using the orthogonality of $U$ and $V$, this simplifies to

$$\Delta W = \sum_{t=1,s=0}^{T-1,t-1} P\bar{W}^s\bar{A}(\bar{A}\bar{W}^t\bar{B} - \bar{A}_\star\bar{W}_\star^{\,t}\bar{B}_\star)\bar{B}\bar{W}^{t-s-1}P^\top, \tag{81}$$

where we also used the assumption that the input, recurrent and output layer sizes are equal, so that $\bar{A}^\top = \bar{A}$ and $\bar{B}^\top = \bar{B}$. Similar derivations yield

$$\Delta B = \sum_{t=0}^{T-1} P\bar{W}^t\bar{A}(\bar{A}\bar{W}^t\bar{B} - \bar{A}_\star\bar{W}_\star^{\,t}\bar{B}_\star)V^\top, \tag{82}$$

$$\Delta A = \sum_{t=0}^{T-1} U(\bar{A}\bar{W}^t\bar{B} - \bar{A}_\star\bar{W}_\star^{\,t}\bar{B}_\star)\bar{B}\bar{W}^t P^\top. \tag{83}$$

**tBPTT**  A similar derivation as for BPTT can be carried out for tBPTT, using expected updates $\Delta_\tau W$ and $\Delta_\tau B$. We obtain

$$\Delta_\tau W = \sum_{t=1,s=0}^{T-1,\min\{\tau-1,t-1\}} P\bar{W}^s\bar{A}(\bar{A}\bar{W}^t\bar{B} - \bar{A}_\star\bar{W}_\star^{\,t}\bar{B}_\star)\bar{B}\bar{W}^{t-s-1}P^\top, \tag{84}$$

$$\Delta_\tau B = \sum_{t=0}^{\tau-1} P\bar{W}^t\bar{A}(\bar{A}\bar{W}^t\bar{B} - \bar{A}_\star\bar{W}_\star^{\,t}\bar{B}_\star)V^\top. \tag{85}$$

**RFLO** The diagonalized RFLO updates are obtained by replacing the leftmost $\bar{W}$ and $\bar{A}$ in the BPTT diagonalization by $\hat{w}\mathbf{I}$ and $\bar{R}$, respectively:

$$
\Delta_{\mathrm{RFLO}}W = \sum_{t=1,s=0}^{T-1,t-1} P\hat{w}^s \bar{R}(\bar{A}\bar{W}^t\bar{B} - \bar{A}_\star \bar{W}_\star^{\,t}\bar{B}_\star)\bar{B}\bar{W}^{t-s-1}P^\top,
$$

$$
\Delta_{\mathrm{RFLO}}B = \sum_{t=0}^{T-1} P\hat{w}^t \bar{R}(\bar{A}\bar{W}^t\bar{B} - \bar{A}_\star \bar{W}_\star^{\,t}\bar{B}_\star)V^\top.
$$

(86)

## E. Asymptotics

We derive the $T \to \infty$ limit of the parameter updates. To do so, we use the following simple properties of geometric series:

$$
\sum_{k=0}^{n-1} r^k = \frac{1-r^n}{1-r} \xrightarrow[n\to\infty]{} \frac{1}{1-r}
$$

(87)

and

$$
\sum_{k=1}^{n-1} r^k = \frac{r-r^n}{1-r} \xrightarrow[n\to\infty]{} \frac{r}{1-r},
$$

(88)

when $|r| < 1$. Also, we need the following for BPTT and tBPTT:

$$
\sum_{k=1}^{n-1} k r^k = \frac{r(1-r^n) - nr^n(1-r)}{(1-r)^2} \xrightarrow[n\to\infty]{} \frac{r}{(1-r)^2},
$$

(89)

when $|r| < 1$. Below, we shall not repeat this convergence condition every time.

Lowercase variables $w$, $a$ and $b$ denote a single mode of $W$, $A$ and $B$, respectively.

### E.1. RFLO

Throughout this subsection, we omit prefactor $\hat{a}$ which appears in the $W$ and $B$ updates. For the recurrent weights $W$, we have

$$
\Delta_{\mathrm{RFLO}}w = \sum_{t=1,s=0}^{T-1,t-1} \hat{w}^s(aw^t b - a_\star w_\star^t b_\star)bw^{t-s-1} = \sum_{t=1,s=0}^{T-1,t-1} (\hat{w}^s ab^2 w^{2t-s-1} - \hat{w}^s a_\star w_\star^t b_\star bw^{t-s-1}) =: S_1 - S_2. \quad (90)
$$

The sum $S_1$ gives

$$
S_1 = ab^2 \sum_{t=1,s=0}^{T-1,t-1} \left(\frac{\hat{w}}{w}\right)^s w^{2t-1} = ab^2 \sum_{t=1}^{T-1} \frac{1-\left(\frac{\hat{w}}{w}\right)^t}{1-\left(\frac{\hat{w}}{w}\right)} w^{2t-1} = \frac{ab^2}{w-\hat{w}} \sum_{t=1}^{T-1} (w^{2t} - (w\hat{w})^t). \quad (91)
$$

As $T \to \infty$,

$$
S_1 \to \frac{ab^2}{w-\hat{w}}\left(\frac{w^2}{1-w^2} - \frac{w\hat{w}}{1-w\hat{w}}\right),
$$

(92)

when $|w| < 1$ and $|w\hat{w}| < 1$. The sum $S_2$ gives

$$
S_2 = a_\star b_\star b \sum_{t=1,s=0}^{T-1,t-1} \left(\frac{\hat{w}}{w}\right)^s (w_\star w)^t w^{-1} = \frac{a_\star b_\star b}{w-\hat{w}} \sum_{t=1}^{T-1} ((w_\star w)^t - (w_\star \hat{w})^t) \to \frac{a_\star b_\star b}{w-\hat{w}}\left(\frac{w_\star w}{1-w_\star w} - \frac{w_\star \hat{w}}{1-w_\star \hat{w}}\right), \quad (93)
$$

when $|w_\star w| < 1$ and $|w_\star \hat{w}| < 1$. Altogether,

$$
\Delta_{\mathrm{RFLO}}w \to \frac{ab^2}{w-\hat{w}}\left(\frac{w^2}{1-w^2} - \frac{w\hat{w}}{1-w\hat{w}}\right) - \frac{a_\star b_\star b}{w-\hat{w}}\left(\frac{w_\star w}{1-w_\star w} - \frac{w_\star \hat{w}}{1-w_\star \hat{w}}\right),
$$

(94)

which is, up to algebraic simplification, the same result as in the main text. The large $T$ limits for the input and output weights follow directly from the same geometric series identities.

### E.2. BPTT

For a single recurrent mode,

$$\Delta w = \sum_{t=1,s=0}^{T-1,t-1} w^s a(aw^t b - a_\star w_\star^t b_\star)bw^{t-s-1}. \tag{95}$$

$$\tag{96}$$

After straightforward algebraic steps, this can be written

$$\Delta w = w^{-1}a^2b^2 \sum_{t=1}^{T-1} tw^{2t} - aa_\star b_\star bw^{-1}\sum_{t=1}^{T-1} t(w_\star w)^t. \tag{97}$$

In the limit $T \to \infty$, using Eq. 89,

$$\Delta w \to a^2b^2 \frac{w}{(1-w^2)^2} - aa_\star b_\star b \frac{w_\star}{(1-w_\star w)^2}, \tag{98}$$

as long as $|w| < 1$ and $|w_\star w| < 1$. The expression for $\Delta b$ is obtained in a similar fashion.

### E.3. tBPTT

We start with

$$\Delta_\tau w = \sum_{t=1,s=0}^{T-1,\min\{\tau-1,t-1\}} a(aw^t b - a_\star w_\star^t b_\star)bw^{t-1}. \tag{99}$$

The sum of $s$ yields $\min\{\tau, t\}$, which separates the sum over $t$ into two parts:

$$\Delta_\tau w = abw^{-1}\left(\sum_{t=1}^{\tau} t[ab(w^2)^t - a_\star b_\star(w_\star w)^t] + \tau \sum_{t=\tau+1}^{T-1} [ab(w^2)^t - a_\star b_\star(w_\star w)^t]\right) \tag{100}$$

$$= abw^{-1}\left(ab\left[\sum_{t=1}^{\tau} t(w^2)^t + \tau \sum_{t=\tau+1}^{T-1} (w^2)^t\right] - a_\star b_\star\left[\sum_{t=1}^{\tau} t(w_\star w)^t + \sum_{t=\tau+1}^{T-1} (w_\star w)^t\right]\right) \tag{101}$$

We have terms of the form $\sum_{t=1}^{\tau} tr^t + \tau \sum_{t=\tau+1}^{T-1} r^t$ where (Eq. 89)

$$\sum_{t=1}^{\tau} tr^t = \frac{r(1-r^{\tau+1}) - (\tau+1)r^{\tau+1}(1-r)}{(1-r)^2} \tag{102}$$

and

$$\tau \sum_{t=\tau+1}^{T-1} r^t = \tau \sum_{t=0}^{T-1} r^t - \tau \sum_{t=0}^{\tau} r^t = \tau \frac{r^{\tau+1} - r^T}{1-r} \longrightarrow \tau \frac{r^{\tau+1}}{1-r}. \tag{103}$$

After some algebra, we get

$$\sum_{t=1}^{\tau} tr^t + \tau \sum_{t=\tau+1}^{T-1} r^t \to \frac{r - r^{\tau+1}}{(1-r)^2}. \tag{104}$$

Therefore,

$$\Delta_\tau w \to abw^{-1}\left(ab\frac{w^2 - (w^2)^{\tau+1}}{(1-w^2)^2} - a_\star b_\star \frac{w_\star w - (w_\star w)^{\tau+1}}{(1-w_\star w)^2}\right), \tag{105}$$

which corresponds to the expression in the main text when $\tau = 1$.

For $b$, we have

$$\Delta_\tau b = \sum_{t=0}^{\tau-1} a(a(w^2)^t b - a_\star(ww_\star)^t b_\star) = a^2 b \frac{1 - (w^2)^\tau}{1 - w^2} + aa_\star b_\star \frac{1 - (ww_\star)^\tau}{1 - ww_\star}, \tag{106}$$

which again corresponds to the expression in the main text when $\tau = 1$.

## F. Stability

We first write the Jacobian matrices at a generic point $(a, b, w)$ for each algorithm. Each ODE system has the form $[\dot{a}, \dot{b}, \dot{w}]^\top = [f_a(a, b, w), f_b(a, b, w), f_w(a, b, w)]^\top$. As usual, the Jacobian is the matrix of partial derivatives of the vector field; for instance, the first row is $[\partial_a f_a, \partial_b f_a, \partial_w f_a]$. Everywhere, we use the diagonalized expected updates in the limit $T \to \infty$.

**tBPTT ($\tau = 1$)** From Eq. 22 and Eq. 20 (for $a$), we have

$$\begin{bmatrix} \dot{a} \\ \dot{b} \\ \dot{w} \end{bmatrix} = - \begin{bmatrix} \frac{ab^2}{1-w^2} - \frac{a_\star bb_\star}{1-ww_\star} \\ a^2 b - aa_\star b_\star \\ \frac{a^2 b^2 w}{1-w^2} - \frac{aa_\star bb_\star w_\star}{1-ww_\star} \end{bmatrix} \tag{107}$$

and the Jacobian matrix is

$$\mathcal{J}_1 = - \begin{bmatrix} \frac{b^2}{1-w^2} & \frac{2ab}{1-w^2} - \frac{a_\star b_\star}{1-ww_\star} & \frac{2ab^2 w}{(1-w^2)^2} - \frac{ba_\star b_\star w_\star}{(1-ww_\star)^2} \\ 2ab - a_\star b_\star & a^2 & 0 \\ \frac{2ab^2 w}{1-w^2} - \frac{ba_\star b_\star w_\star}{1-ww_\star} & \frac{2a^2 bw}{1-w^2} - \frac{aa_\star b_\star w_\star}{1-ww_\star} & \frac{a^2 b^2}{1-w^2} + \frac{2a^2 b^2 w^2}{(1-w^2)^2} - \frac{aba_\star b_\star w_\star^2}{(1-ww_\star)^2} \end{bmatrix}. \tag{108}$$

Note the minus signs in front of both matrices above.

We evaluate the Jacobian on the line of equilibria $ab = a_\star b_\star$, $w = w_\star$ using the parametrization $(a, b, w) = (s, a_\star b_\star s^{-1}, w_\star)$ to get

$$\mathcal{J}_1 = - \begin{bmatrix} a_\star^2 b_\star^2 s^{-2} m_\star & a_\star b_\star m_\star & a_\star^2 b_\star^2 s^{-1} w_\star m_\star^2 \\ a_\star b_\star & s^2 & 0 \\ a_\star^2 b_\star^2 s^{-1} w_\star m_\star & sa_\star b_\star w_\star m_\star & a_\star^2 b_\star^2 m_\star^2 \end{bmatrix}, \tag{109}$$

using the definition

$$m_\star = \frac{1}{1 - w_\star^2}. \tag{110}$$

The characteristic polynomial is

$$P(\lambda) = \lambda^3 + \lambda^2 (s^2 + a_\star^2 b_\star^2 s^{-2} m_\star + a_\star^2 b_\star^2 m_\star^2) + \lambda(s^2 a_\star^2 b_\star^2 m_\star^2 + a_\star^4 b_\star^4 s^{-2} m_\star^2). \tag{111}$$

On the line $a = b = 0$ and $w$ free, we get

$$\mathcal{J}_1 = \begin{bmatrix} 0 & \frac{a_\star b_\star}{1-ww_\star} & 0 \\ a_\star b_\star & 0 & 0 \\ 0 & 0 & 0 \end{bmatrix}, \tag{112}$$

yielding $P(\lambda) = \lambda \left( \lambda^2 - \frac{a_\star^2 b_\star^2}{(1-ww_\star)} \right)$.

**RFLO** From Eq. 20, we have

$$\begin{bmatrix} \dot{a} \\ \dot{b} \\ \dot{w} \end{bmatrix} = - \begin{bmatrix} \frac{ab^2}{1-w^2} - \frac{a_\star bb_\star}{1-ww_\star} \\ \frac{\hat{a}ab}{1-\hat{w}w} - \frac{\hat{a}a_\star b_\star}{1-\hat{w}w_\star} \\ \frac{\hat{a}ab^2 w}{(1-\hat{w}w)(1-w^2)} - \frac{\hat{a}a_\star bb_\star w_\star}{(1-\hat{w}w_\star)(1-w_\star w)} \end{bmatrix}, \tag{113}$$

and the Jacobian matrix is

$$\mathcal{J}_{\text{RFLO}} = - \begin{bmatrix} \frac{b^2}{1-w^2} & \frac{2ab}{1-w^2} - \frac{a_\star b_\star}{1-ww_\star} & \frac{2ab^2 w}{(1-w^2)^2} - \frac{ba_\star b_\star w_\star}{(1-ww_\star)^2} \\ \frac{\hat{a}b}{1-\hat{w}w} & \frac{\hat{a}a}{1-\hat{w}w} & \frac{\hat{a}ab\hat{w}}{(1-\hat{w}w)^2} \\ \frac{\hat{a}b^2 w}{(1-\hat{w}w)(1-w^2)} & \frac{2\hat{a}abw}{(1-\hat{w}w)(1-w^2)} - \frac{\hat{a}a_\star b_\star w_\star}{(1-\hat{w}w_\star)(1-ww_\star)} & \frac{\hat{a}ab^2(1+w^2-2\hat{w}w^3)}{(1-\hat{w}w)^2(1-w^2)^2} - \frac{\hat{a}a_\star bb_\star w^2_\star}{(1-\hat{w}w_\star)(1-w_\star w)^2} \end{bmatrix}. \tag{114}$$

The evaluation on $ab = a_\star b_\star$, $w = w_\star$, using the same parametrization as above, yields

$$\mathcal{J}_{\text{RFLO}} = - \begin{bmatrix} a_\star^2 b_\star^2 s^{-2} m_\star & a_\star b_\star m_\star & a_\star^2 b_\star^2 s^{-1} w_\star m_\star^2 \\ \hat{a}a_\star b_\star s^{-1}\hat{m}_\star & \hat{a}s\hat{m}_\star & \hat{a}a_\star b_\star \hat{w}\hat{m}_\star^2 \\ \hat{a}a_\star^2 b_\star^2 s^{-2} w_\star \hat{m}_\star m_\star & \hat{a}a_\star b_\star w_\star \hat{m}_\star m_\star & \hat{a}a_\star^2 b_\star^2 s^{-1}(1-\hat{w}w_\star^3)\hat{m}_\star^2 m_\star^2 \end{bmatrix}, \tag{115}$$

using the definition

$$\hat{m}_\star = \frac{1}{1-\hat{w}w_\star}. \tag{116}$$

The characteristic polynomial is

$$P(\lambda) = \lambda^3 + (a_\star^2 b_\star^2 s^{-2} m_\star + \hat{a}s\hat{m}_\star + \hat{a}a_\star^2 b_\star^2 s^{-1}(1-\hat{w}w_\star^3)\hat{m}_\star^2 m_\star^2)\lambda^2 \tag{117}$$

$$+ (\hat{a}^2 a_\star^2 b_\star^2 + \hat{a}a_\star^4 b_\star^4 s^{-3})m_\star^2 \hat{m}_\star^2 \lambda. \tag{118}$$

**BPTT**  From Eq. 21, we have

$$\begin{bmatrix} \dot{a} \\ \dot{b} \\ \dot{w} \end{bmatrix} = - \begin{bmatrix} \frac{ab^2}{1-w^2} - \frac{a_\star bb_\star}{1-ww_\star} \\ \frac{a^2 b}{1-w^2} - \frac{aa_\star b_\star}{1-ww_\star} \\ \frac{a^2 b^2 w}{(1-w^2)^2} - \frac{aa_\star bb_\star w_\star}{(1-ww_\star)^2} \end{bmatrix} \tag{119}$$

$$\mathcal{J}_{\text{bptt}} = - \begin{bmatrix} \frac{b^2}{1-w^2} & \frac{2ab}{1-w^2} - \frac{a_\star b_\star}{1-ww_\star} & \frac{2ab^2 w}{(1-w^2)^2} - \frac{ba_\star b_\star w_\star}{(1-ww_\star)^2} \\ \frac{2ab}{1-w^2} - \frac{a_\star b_\star}{1-ww_\star} & \frac{a^2}{1-w^2} & \frac{2a^2 bw}{(1-w^2)^2} - \frac{aa_\star b_\star w_\star}{(1-ww_\star)^2} \\ \frac{2ab^2 w}{(1-w^2)^2} - \frac{a_\star bb_\star w_\star}{(1-ww_\star)^2} & \frac{2a^2 bw}{(1-w^2)^2} - \frac{aa_\star b_\star w_\star}{(1-ww_\star)^2} & \frac{a^2 b^2(1+3w^2)}{(1-w^2)^3} - \frac{2aa_\star bb_\star w^2_\star}{(1-ww_\star)^3} \end{bmatrix}. \tag{120}$$

On $a = b = 0$ and free $w$:

$$\mathcal{J}_{\text{bptt}} = - \begin{bmatrix} 0 & \frac{a_\star b_\star}{1-ww_\star} & 0 \\ \frac{a_\star b_\star}{1-ww_\star} & 0 & 0 \\ 0 & 0 & 0 \end{bmatrix}. \tag{121}$$

The characteristic polynomial is $\lambda\left(\lambda^2 - \frac{a_\star^2 b_\star^2}{(1-ww_\star)^2}\right)$ with roots $\lambda_0 = 0$, $\lambda_\pm = \pm a_\star b_\star /|1-ww_\star|$.

On the optimal manifold,

$$\mathcal{J}_{\text{bptt}} = - \begin{bmatrix} a_\star^2 b_\star^2 s^{-2} m_\star & a_\star b_\star m_\star & a_\star^2 b_\star^2 s^{-1} w_\star m_\star^2 \\ a_\star b_\star m_\star & s^2 m_\star & sa_\star b_\star w_\star m_\star^2 \\ a_\star^2 b_\star^2 s^{-1} w_\star m_\star^2 & sa_\star b_\star w_\star m_\star^2 & a_\star^2 b_\star^2 m_\star^3 (1+w_\star^2) \end{bmatrix}, \tag{122}$$

yielding

$$P(\lambda) = \lambda^3 + [a_\star^2 b_\star^2 s^{-2} m_\star + s^2 m_\star + a_\star^2 b_\star^2 m_\star^3(1+w_\star^2)]\lambda^2 + (s^2 a_\star^2 b_\star^2 + a_\star^4 b_\star^4 s^{-2})m_\star^4 w_\star^2 \lambda. \tag{123}$$

## G. Low-Rank Structure of RFLO Learning

In this section we prove 3.1. This result follows straightforwardly from Eq. 9. Using this equation, along with the update equation $\theta_{k+1} = \theta_k - \eta\Delta\theta_k$, and writing a learned solution of RFLO, $W_K$, as a sum over $K$ gradient estimate updates, we get:

$$W_K = W_0 - \eta \sum_{k=1}^{K} \sum_{t=1,s=0}^{T-1,t-1} [R\widehat{W}^s]^\top \mathcal{E}_t^k [W_k^{t-s-1} B_k]^\top \tag{124}$$

$$= W_0 - R^\top \sum_{k,t,s} \eta\hat{w}^s \mathcal{E}_t^k B_k^\top W_k^{t-s-1}{}^\top \tag{125}$$

$$= W_0 + R^\top Q, \tag{126}$$

where we defined $Q = -\sum_{k,t,s} \eta \hat{w}^s \mathcal{E}_t^k B_k^\top W_k^{t-s-1}{}^\top$. Recall that $R^\top \in \mathbb{R}^{n \times o}$, and observe that $Q \in \mathbb{R}^{o \times n}$. This proves the result for the matrix $W$. To show that RFLO-learned values of $B$ are also rank-$o$ perturbations of the initialization of $B$, one proceeds similarly. Starting now with the expression for $B$ analogous to Eq. 9, and working through steps analogous to those above, one arrives at the result given in 3.1, where $q_i^{(b)}$ are row vectors of the matrix $Q^{(b)} = -\sum_{k,t,s} \eta \hat{w}^s \mathcal{E}_t^k$.

## H. Generalizing to a Sequence Loss

For ease of exposition we work in the main text with the loss $L_T = \frac{1}{2}||y_T - y_T^\star||^2$, but we can generalize our main results straight-forwardly to the sequence loss $\mathcal{L} = \frac{1}{2N} \sum_{T=1}^N ||y_T - y_T^\star||^2$.

For BPTT on the sequence loss, because of linearity of expectation and derivative we get the following (where, as in the main text, we assume that we can interchange differentiation and expectation):

$$\nabla_\theta \mathcal{L} = \frac{1}{N} \sum_{T=1}^N \nabla_\theta L_T. \tag{127}$$

For RFLO and tBPTT, moving to the sequence loss causes these algorithms to no longer be local, as we now have to save the learning update at each $T$ before applying it in the sum. However, it is still interesting to consider the sequence loss for these algorithms if we simply view the averaging over the length $N$ sequence as coming about through a low-pass filtering of individual time-point updates. We thus see that the sequence-loss update for each algorithm, for parameter $\theta$, has the following general sample mean-type form:

$$\Delta_\mathcal{L} \theta = \frac{1}{N} \sum_{T=1}^N \Delta(\theta, T), \tag{128}$$

where $\Delta(\theta, T)$ is the update according to the loss $L_T$ for the given parameter and algorithm as in the main body of the paper.

Our main results from sections 3.1, 3.2, and 3.3 imply that an arbitrary element of the update array, $\Delta(\theta, T)$, converges to a value $\Delta\theta$ as $T \to \infty$. We can use this to show that, in the large $N$ limit, the sequence loss, $\mathcal{L}$, actually gives exactly the same element-wise updates as the large-$T$ limit of the point-wise loss, $L_T$, used in the body of the paper:

$$\lim_{N \to \infty} \Delta_\mathcal{L} \theta = \lim_{N \to \infty} \frac{1}{N} \sum_{T=1}^N \Delta(\theta, T) \tag{129}$$

$$= \lim_{N \to \infty} \frac{1}{N} \sum_{T=1}^N \Delta\theta + \frac{1}{N} \sum_{T=1}^N [\Delta(\theta, T) - \Delta\theta] \tag{130}$$

$$= \Delta\theta + \lim_{N \to \infty} \frac{1}{N} \sum_{T=1}^N [\Delta(\theta, T) - \Delta\theta]. \tag{131}$$

Let $\epsilon > 0$ be arbitrary. Because of the convergence of $\Delta(\theta, T)$ we can find and fix $S$ such that, for all $T \geq S$, $|\Delta(\theta, T) - \Delta\theta| < \frac{\epsilon}{2}$. Then

$$\left| \frac{1}{N} \sum_{T=1}^N [\Delta(\theta, T) - \Delta\theta] \right| \leq \frac{1}{N} \sum_{T=1}^S |\Delta(\theta, T) - \Delta\theta| + \frac{1}{N} \sum_{T=S+1}^N |\Delta(\theta, T) - \Delta\theta| \leq \frac{1}{N} C_S + \frac{\epsilon}{2}, \tag{132}$$

where $C_S$ is a constant that does not depend on $N$. Selecting $N_0 \geq \frac{2C_S}{\epsilon}$ shows that we can find $N_0$ s.t., $\forall N \geq N_0$, $\frac{C_S}{N} \leq \frac{\epsilon}{2}$, thus demonstrating that the second term in Eq. 131 converges to zero, by definition, and completing the generalization of our results on data-aligned networks to the sequence loss.

Lastly, the low rank result of section 3.6 can be easily seen to apply almost without change to the sequence case. The only difference is that there is now an extra sum over $T$ between the sum over $k$ and the sums over $t$ and $s$ in Equation 125, and, if we consider the mean, an added factor of $N^{-1}$. This extra sum and factor then show up accordingly in $Q$ and $Q^{(b)}$.

# I. Experimental Methods

## I.1. Methods for Fig. 5

We used parameters:

- $n = 10$, $n_\star = 4$, $m = m_\star = 4$, $o_\star = o = 4$.

- $W_\star$: diagonal matrix with eigenvalues: 0.9, 0.7, 0.5, 0.3.

- $A \sim \mathcal{N}(0, \sqrt{2/n}/100)$, $B \sim \mathcal{N}(0, \sqrt{2/m}/100)$, $W \sim \mathcal{N}(0, \sqrt{2/(n+m)}/100)$, and, for RFLO, $\hat{w} = 1.0$ and $R \sim \mathcal{N}(0, \sqrt{2/n}/100)$.

- $A_\star = B_\star = \mathbf{I}_4$, the identity matrix in $\mathbb{R}^{4 \times 4}$.

The student network was trained on 750 samples of length 100 and the performance was evaluated on 1 held out sample of length 100. Batch size was 750. Training was run for $35,000$ epochs, or until the loss reached a value of $10^{-8}$ (whichever came first), using SGD with default Pytorch parameters. Learning rates of $0.005$ were used. After calculating the matrix decomposition of $W$ in this experiment we discarded the imaginary parts of all eigenvalues/vectors. This is because we found they did not contribute much to the learning process, ostensibly given the lack of rotational dynamics in the teacher.

## I.2. Methods for Fig. 6

For this experiment, we used parameters:

- $n = 20$, $n_\star = 10$, $m = m_\star = 10$, $o_\star = o = 1$.

- $W_\star$: diagonal matrix with eigenvalues: 0.8280, 0.8280, 0.7360, 0.7360, 0.6440, 0.6440, 0.6440, 0.5520, 0.5520, 0.5520.

- $A_\star \sim \mathcal{N}(0, \sqrt{2/n_\star})$, $B_\star \sim \mathcal{N}(0, \sqrt{2/m_\star})$, $A, R \sim \mathcal{N}(0, \sqrt{2/n})$, $B \sim \mathcal{N}(0, 2/m)$, $W \sim \mathcal{N}(0, \sqrt{2/(n+m)})$, and $\hat{w} = 0.8$

The student network was trained on 200 samples of length 150 and the performance was evaluated on 50 held out samples of length 150. Unlike the theory, the loss was evaluated at every timestep. Training was run for 1500 epochs with batch sizes of 200 (i.e. 1500 iterations of gradient descent), using Adam, with gradients clipped to 0.1. Learning rates of $0.00005$ were used for all algorithms. Initial conditions that led to divergence we discarded, where networks were considered diverged if their power level (validation error) was over 9000.

## I.3. Methods for Supplementary Figures

We omit mention of figures whose methods are entirely described by previously outlined experimental methods and/or in their figure captions.

### I.3.1. FIGURES 8 AND 9

Same as the methods for Fig. 5 except for two differences. First, these experiments were only run for 10000 epochs or until the convergence criterion. Second, the experiment with bptt was run with training and validation sequence lengths of 200.

## I.4. Code Availability

Code to generate all figures is available here: `https://gitlab.com/zek3r/icml-2026`

# J. Supplementary Figures

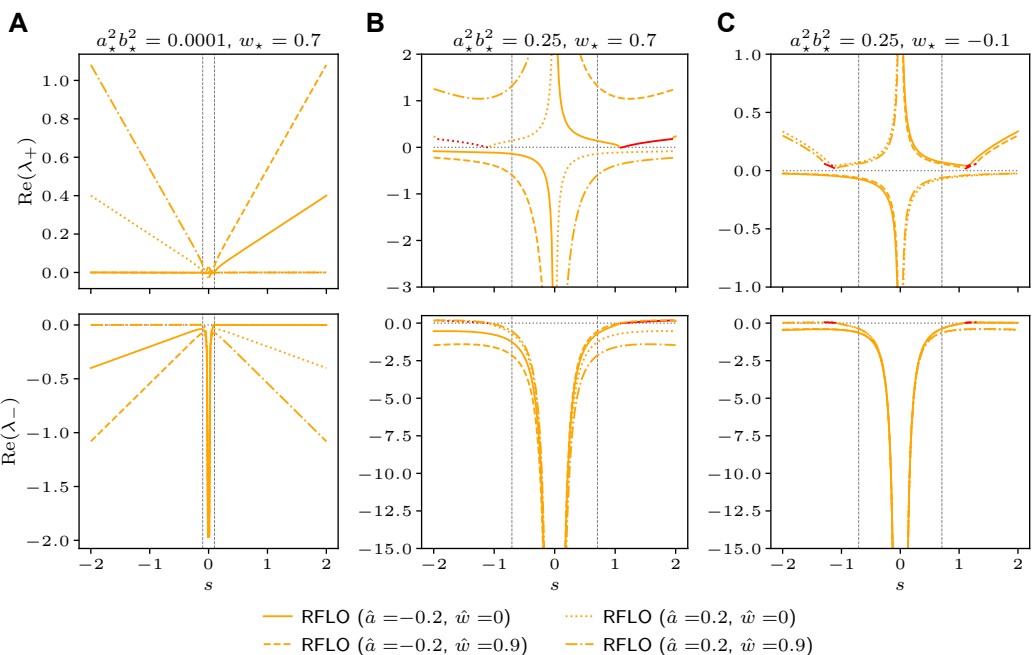

*Figure 7.* Same as Fig. 3 in the main text, but comparing low ($\hat{w} = 0$) and high ($\hat{w} = 0.9$) values for $\hat{w}$ for the RFLO learning rule.

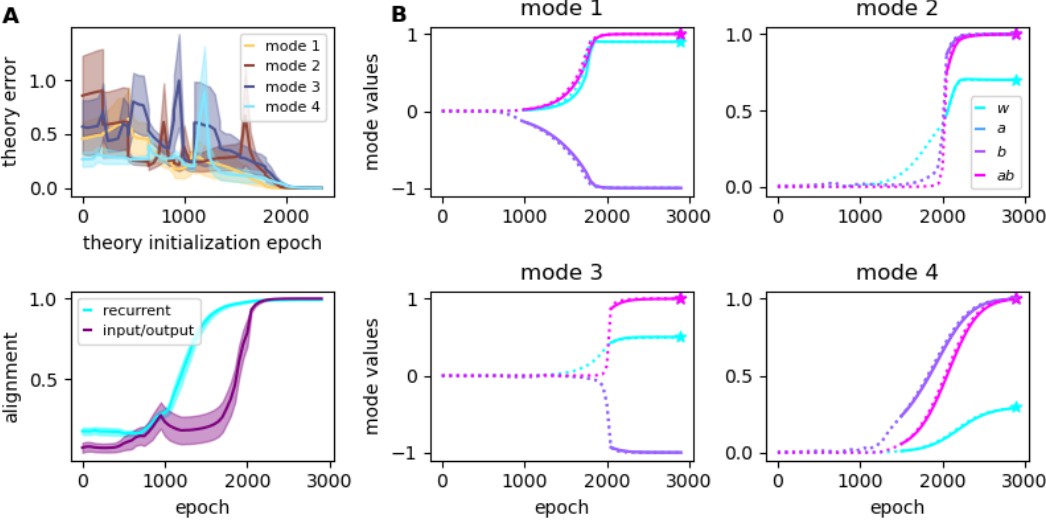

*Figure 8.* As in Fig. 5 but for BPTT.

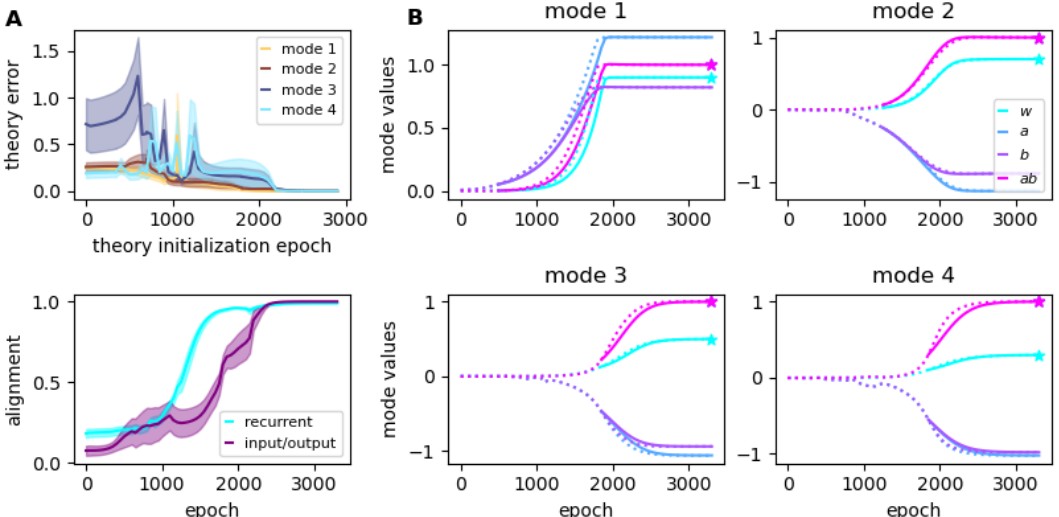

*Figure 9.* As in Fig. 5 but for tBPTT.

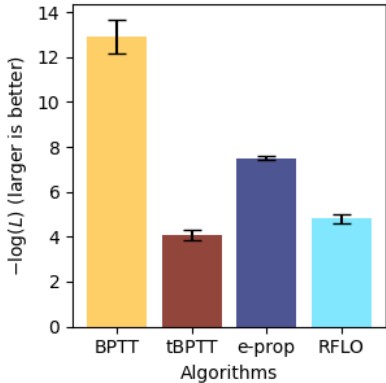

*Figure 10.* Negative log error of the four algorithms on the task from Section 3.6. RFLO and tBPTT both find low rank solutions and have lower performance; BPTT finds the highest rank solution and sees best performance, and e-prop is intermediate on both counts.

