# OpenReview forum: "Dynamics and Representation Structure of Local Approximations to Gradient-Based Learning in Linear Recurrent Neural Networks"
_ICML.cc/2026/Conference — ICML 2026 regular_

### Official Review · Reviewer_NdRD · 2026-03-13

**Soundness:** 3
**Presentation:** 2
**Significance:** 2
**Originality:** 2
**Overall Recommendation:** 4
**Confidence:** 3

**Summary:**

This paper studies the training dynamics of linear RNN in a specific data-aligned teacher-student setting of three different common algorithms: backward propagation through time(BPTT), truncated backward propagation through time (tBPTT), and random feedback local online(RFLO). The authors characterize the training dynamics of the algorithms, deriving the ODE and the stationary solutions manifold in this specific setting. Based on these theoretical insights, the paper conducts mechanistic experiments and theoretical analysis to show the stability properties and convergence rate of different algorithms. The paper basically finds that BPTT has some similar fixed-point properties, while RFLO exhibits a different low-rank bias.

**Compliance With Llm Reviewing Policy:**

Affirmed.

**Final Justification:**

The authors almost fully resolved my concern so I increased the score. However, the current paper is still kind of confusing, so I recommend the authors improve the writing in the next revision of this paper. For example, "However, surprisingly, for s values very close to 0, RFLO achieves the fastest convergence rate." in Line 269-271 right column is distracting and made me think RFLO may even 'have better convergence rate' in certain cases. And thus I cannot strongly recommend accepting this paper.

**Key Questions For Authors:**

- How generalizable is the theoretical setting? What if we lift the data alignment assumption? Also can those theoretical implication apply to at least slightly loosened assumptions (e.g.  not data aligned but randomly initialized teacher-student setting)?
- Can you stress more on the main contribution of this paper in the introduction?
- Why could these insights apply to SSMs? Can you do more experiments?

**Limitations:**

yes

**Strengths And Weaknesses:**

Strength:
- The paper discusses the training dynamics of linear RNNs, which are related to recent popular state space models. The topic is pretty relevant and analyzing the algorithms, especially understanding the properties of tBPTT, is quite interesting.
- The paper combines manifold interpretability experiments with rigorous analysis under a specific theoretical setting, where the dynamics can be explicitly characterized. The analysis on the dynamics is close to end-to-end and detailed. It explicitly shows the bias of RFLO and the similarity between BPTT and tBPTT, which could be another novel contribution.

Weakness:
- The contribution is not stressed in the introduction clearly, and it is a bit confusing to get the main message. There are many components and messages in the analysis, but it seems hard to interpret why those analyses and conclusions are connected to the training of real linear RNNs. I would recommend that the authors discuss more of their contributions to the community.
- The analysis setting seems too specific, which cannot generally represent SSMs and linear RNNs. In this setting with data aligned assumption, each singular vector dimension is decoupled and the analysis soon becomes one-dimensional (since all other dimensions are independent), which could be an oversimplification. Moreover, RFLO may even have better convergence rate and better landscape, which is quite counterintuitive and cannot be supported through observations real-world experiments. That may also be invalid in real-world experiments. The paper generally lacks larger-scale experiments showing the applicability of those results. So the claim by the authors, "Our work provides fundamental insights into how locality constraints shape RNN learning dynamics, with implications for neuroscientific models of learning and alternative optimization approaches for state-space models," seems a bit ungrounded.

---

> ### Author Rebuttal · Authors · 2026-03-31
>
> Thank you very much for the detailed and thoughtful review! We believe that the changes that you inspired have improved our manuscript. We respond to your comments below.
>
> ## Weaknesses
> 1. Thank you for the helpful advice. We have tried to write out the specific mathematical contributions in points (1)-(4) of our paper, but we realize that our communication of the value of these contributions perhaps lacked clarity. To this end, we  modify the introduction to provide deeper practical intuition on what we view as our primary two contributions: (1) building understanding of learning dynamics for local-learning algorithms in a particular learning regime—i.e. “data-aligned”; (2) showing that the popular algorithm RFLO will necessarily learn low-rank structure in linear networks, regardless of whether the networks are data-aligned or not. The first point is valuable because the field currently has very limited knowledge of gradient-learning dynamics (see response to weakness 1 of Reviewer XwWR), let alone local approximations to it. Our work shows that, in the data-aligned case, RFLO and tBPTT will learn optimal solutions. This seems valuable because past work has studied this data-aligned regime with BPTT alone (see ICML spotlight by Proca et al. 2025) and our study provides more context for these results, and because we believe that this regime can help us understand the learning process for RNNs when the data being learned has sufficiently independent dynamic modes. The second point is valuable because low-rank structure is of interest in machine learning (see e.g. Low Rank Adaptation), and neuroscience (see work by Srdjan Ostojic et al.), and because recognizing that RFLO is limited to learning low-rank perturbations in linear networks puts hard constraints on what it can and cannot learn. Please let us know if this update will appropriately address your concern.
> 2. We wish to respond to three points here in particular. First, we agree that the setting is limiting, however as we have outlined in our response to weakness 1 of Reviewer XwWR, there is very limited existing theoretical literature. The work that has been done has typically used assumptions quite similar to ours and has still found that the resultant intuition has generalized in useful ways to RNNs and SSMs alike, even, in some cases, to nonlinear RNNs. We also note that our results on the low-rank learned structure of RFLO do not require data-aligned assumptions.
> Second, we are curious about what you mean by RFLO having a better convergence rate and better landscape. With the assumptions we make, we indeed find that RFLO will typically learn slower than tBPTT or BPTT alike.
> Third, we recognize that whether or not a result is “fundamental” is subjective, so we will remove this wording in the manuscript.
> ## Questions
> 1. This is a fantastic question. In light of previous work on alignment in feed-forward neural networks (see [Atanasov et al. 2021](https://arxiv.org/abs/2111.00034)), and some preliminary experiments that we have run, we believe that these results should generalize to describe learning even when the trained RNN is not data aligned, under particular conditions. We expect the important conditions to be that the data the student is learning can be represented by sufficiently decoupled computational modes (corresponding at least approximately to the assumptions that we make on the teacher in our study), and that the student has undergone sufficient initial training such that it is now in a quasi-data aligned regime. We view this as an important direction for future research.
> Lastly, we wish to emphasize that the results of 3.4 do not require the data-aligned assumption at all.
> 2. Yes, thank you for this point. See our response to weakness #1 of your review.
> 3. Thank you for the question. First, as the results of section 3.4 do not rely on data-alignment and instead apply to linear RNNs, we believe that they should provide useful intuition for the recurrent layer of single-layer SSMs given their linear nature. Second, while the nonlinear input/output maps can complicate the picture, previous work has observed that theory from diagonal networks can still provide valuable insights for learning dynamics in SSMs and RNNs more broadly (see references 4 and 5 in response to weakness 1 of Reviewer XwWR).

---

> > ### Author Rebuttal · Reviewer_NdRD · 2026-04-03
> >
> > Thank you for the detailed response. I increased my score to 4.

---

> > > ### Author Response · Authors · 2026-04-08
> > >
> > > Thank you very much for working with us to improve our manuscript, and for the updated score! We wish to note that, on the request of another reviewer, we have now generalized all our results from a single-time point loss to a sequence loss (see response to reviewer zH6L). Please let us know if there is anything else we can do that would encourage you to further support our paper, and increase the likelihood that it is made available to the community. Thanks again

---

### Official Review · Reviewer_pLBe · 2026-03-13

**Soundness:** 4
**Presentation:** 4
**Significance:** 3
**Originality:** 3
**Overall Recommendation:** 5
**Confidence:** 4

**Summary:**

The authors take the linear RNN and consider the training dynamics of backpropagation through time (BPTT), truncated backpropagation through time (tBPTT), and random feedback local online (RFLO) learning. They use a teacher-student setup in the ODE limit, in conjunction with the data-alignment assumption, used in other studies (like Saxe et al. and Proca et al.). This leaves them with a tractable set of 'decoupled' ODE's for singular values of the $W$ matrix, simplifying the analysis of the fixed point manifolds. Authors find BPTT and one-step tBPTT to be similar, while RFLO is qualitatively distinct. They also argue that in general RFLO provides low-rank perturbations to the initial parameters.

**Compliance With Llm Reviewing Policy:**

Affirmed.

**Final Justification:**

I am staying with my evaluation. I see some other reviewers have increased their score.

**Key Questions For Authors:**

I usually have some questions for most authors, but, in this case, I found their exposition to be very clear and useful.

**Limitations:**

Yes.

**Strengths And Weaknesses:**

Strengths: This is a clean analysis comparing two approximate algorithms to BPTT, in a 'time-honored' setting. While many ingredients of the approach existed before, the results seem novel. The authors also relaxed the data-alignment assumption for the analysis of the low-rank nature of the RFLO updates. The study of these local approximations is important for understanding bio-plausible realizations of recurrent neural networks.

Weakness: Of course, one could complain about the use of the data alignment assumption. The authors suggest some ways of going beyond that.

---

> ### Author Rebuttal · Authors · 2026-03-31
>
> Thank you very much for the endorsement—we are happy that you seem as excited about this work as we are! Please let us know if there are any changes we should make that might lead you to consider increasing your rating of our paper.

---

> > ### Author Rebuttal · Reviewer_pLBe · 2026-04-03
> >
> > Thanks for your comments. I am enthusiastic about the work, even if it is in the context of a linear model. I have to see how the other reviewers respond to the respective rebuttals. I am currently sticking to my score.

---

> > > ### Author Response · Authors · 2026-04-08
> > >
> > > Understood. We note for your consideration that, at the time of writing this, two of the other reviewers have now increased their scores to a 3 and a 4 respectively. Lastly, we also note that, on the request of another reviewer, we have generalized all our results from a single-time point loss to a sequence loss (see response to reviewer zH6L). We thank you again for your review, for your time, and for your support of our paper.

---

### Official Review · Reviewer_zH6L · 2026-03-13

**Soundness:** 2
**Presentation:** 1
**Significance:** 2
**Originality:** 2
**Overall Recommendation:** 3
**Confidence:** 4

**Summary:**

The paper studies the learning dynamics of local learning rules and compares them to gradient learning dynamics (backpropagation through time - BPTT) in recurrent neural networks (RNNs). In particular, the authors focus on three rules: exact BPTT, one-step truncated BPTT (tBPTT), and random feedback local online learning (RFLO). They employ a teacher - student setting of data-aligned networks , where under the assumption of whitened inputs and symmetric real recurrent matrices, the learning dynamics can be reduced to mode-wise equations (following the work of Saxe et al.) and be compared in terms of fixed points, stability, and convergence.
The main finding of the paper is that one-step tBPTT remains much closer to BPTT than RFLO does, while RFLO induces qualitatively different learning dynamics because it removes nonlocal gradient terms and no longer optimizes a well-defined objective.


Overall the paper engages with a highly important emerging topic, providing initial steps for a mechanistic explanation of how local learning rules shape learning dynamics and representations in RNNs. However, precisely for the reason of the importance of the topic, the current level of development feels insufficient.  Rather than reading as a mature contribution,the manuscript comes across as a somewhat hurried attempt to plant a flag in a promising area.  The paper sketches out an attractive research topic, but the actual execution remains preliminary relative to the significance of the problem. I would have liked to see a more thorough treatment before publication.   I was struck by this especially because the overall research direction and questions, that are superficially tackled in this paper, are close to research programs that have already been articulated elsewhere and thus the manuscript reads more like flag-planting than a mature contribution ready to be published. Thus I would consider that accepting this paper would prematurely legitimize an insufficiently developed treatment of such important research questions, and would therefore neither serve the field well nor benefit the community.



The topic the paper tackles is important, and there are interesting ideas here, but the manuscript gives the strong impression that the authors are trying to claim ownership over the broad question of how local learning rules shape learning dynamics and representations in RNNs without having done thorough conceptual and empirical work. What might have been better developed as two or three carefully prepared and thoroughly studied papers, has instead been compressed into one paper with prose that resembles better a lab notebook than a polished work that deserves to be published in a high impact venue. I recommend the authors to take as an example the excellent writing style of Saxe et al., and Proca et al., and reflect more on their work and rewrite their results (if not the entire paper).

Overall my perspective is that the paper needs a **major revision** that could then lead to a very impactful paper.

 I detail my various concerns below.

**Compliance With Llm Reviewing Policy:**

Affirmed.

**Final Justification:**

I would like to thank the authors for responding to  the concerns raised by me and the other reviewers.
I appreciate the additional argument in their reply addressing my concern about non-terminal losses, and I am therefore willing to raise my score to weak reject.

That said, my overall assessment remains largely unchanged. I still feel that several of the paper’s results remain undigested, and that the theory has not been stress-tested thoroughly enough. For this reason, my remaining impression is that the manuscript was submitted prematurely. In my view, assigning it an acceptance-level score would not be fair to other papers that were already polished at the time of submission and whose authors also used the rebuttal period to provide substantial additional analyses and rewrites/skeletons or clarifications of sections that were initially difficult to follow. I did not see a comparable level of effort here, either in the original submission or in the rebuttal. I understand that authors may have different time constraints or other commitments during the rebuttal period, but I can only evaluate the manuscript based on the present submission and the authors' responses.

I continue to believe that the paper addresses an interesting topic. Precisely for that reason, I think it would be unfortunate to publish it in its current form, before a number of reasonable questions have been explored more thoroughly, and before the presentation has been brought to the standard the topic deserves. The authors build on two exceptionally well-written papers, Saxe et al. (2014) and Proca et al. (2025), both of which provide strong examples not only in terms of technical development, but also in the structure of the main text and supplement. I believe those works could serve as inspiration as the authors revise and reorganise their manuscript.

**Key Questions For Authors:**

- How essential are the assumption for the claims of similarity between BPTT and one step tBPTT to continue to be valid ( the data-aligned networks, real-symmetric matrices ).



- I think the authors should mention/discuss the following papers: [1], [2], [3]. Probably also this [4].

- "We hope that this work will aid interpretability for RNNs in ML" how? RNNs in ML are usually not trained with local learning rules.

## Comments/Minor

- Please add equation number and streamline the text in the supplement.
- Line 323 the "respectively" is not necessarily needed.


---

### References


[1] Bredenberg, C., Williams, E., Savin, C., Richards, B., & Lajoie, G. (2023). Formalizing locality for normative synaptic plasticity models. Advances in neural information processing systems, 36, 5653-5684

[2] Marschall, O., Cho, K., & Savin, C. (2019). Using local plasticity rules to train recurrent neural networks. arXiv preprint arXiv:1905.12100. (extended abstract)

[3] Marschall, O., Cho, K., & Savin, C. (2020). A unified framework of online learning algorithms for training recurrent neural networks. Journal of machine learning research, 21(135), 1-34.

[4] Marschall, O., & Savin, C. (2023). Probing learning through the lens of changes in circuit dynamics. bioRxiv, 2023-09.

**Limitations:**

Yes

**Strengths And Weaknesses:**

>

## Strengths

- The paper tackles a rather interesting and exciting topic
- The authors employ the necessary simplifying assumptions to make the problem tractable.

- The most interesting result is the proposition 3.1, which says that weight updates learned by RFLO for linear RNNs are
 low-rank.


----
>


## Weaknesses

- The paper both in terms of presentation, numerical evaluation, and writing seems a bit rushed. This results in some places having mathematical inconsistencies.
- The authors exaggerate the generality of their findings without acknowledging the limitations of their assumptions.
- The method assumes whitened inputs, which might be ok to make the problem tractable, but should be explicitly acknowledged in the limitations, while the authors should further probe how far their theoretical results hold when this assumption is violated.
- The presentation of the paper is a bit poor at some points, which goes along the lines with my comment  above that the paper feels rushed.
    - For instance the paper keeps referring to “data-aligned networks” without taking the time to explain what the characteristics of these networks are. To the best of my knowledge, except from a handful of people, no one else in the community understands what is meant with this phrase.
    - The expectation symbol sometimes appears with braces [ ] sometimes not. Also please indicate when necessary what variable/distribution the expectation is taken over.
    - Set of equations before Eq. 12, in the second line missing summation over s?
    - Also please number all your equations both in main but especially in Supplement, where nearly no equation is numbered.
    - There is no boldfacing of vectors.

    - In the results section, the fixed-point manifold subsection and the results subsection are difficult to make sense of. Please take more space to describe this part of results, and separate the methodology ( how you solve the equations ) from what you observe (what the actual results are). In the current version it is difficult for the reader to get the take home message from this subsection. These sections need major restructuring and rewriting. In general the entire results section is difficult to read because the authors keep alternating between methodology, observations and interpretation in a random manner. My take is that they should take some time, reflect on the results and the interpretation, mention only the necessary details about the derivation in the main text, and then draw conclusions. The current version of the manuscript made me seriously annoyed when trying to go through it.
- In some part of the supplement the authors throw equations like knives in a knife-throwing act, for instance on page 18. Please take the time to properly present your derivations and walk the reader through the equations instead of dropping them as is. This very much aligns with my comment above that the prose reads as lab notebook.
    - Figure 2 misses a concrete description. Please indicate either with a legend key or in the caption that the red dots are saddles and blue lines are fixed point manifolds.
    - Throughout the text, the authors introduce the lower case mode-wise singular values  $a$ and $b$, without giving an intuition to the reader what these stand for, while the entire analysis/plots heavily rely on the values of $a$ and $b$. So I would suggest to give them names and to refer in the text to them also with the names (input/output gain? or sth along these lines).
- I find the abbreviation TCA somewhat misleading, since it is commonly associated with tensor component analysis in search results. Please use TempCA or another less ambiguous abbreviation instead. Current abbreviation may lead search results for tensor component analysis (TCA) pointing to your paper.

---

> ### Author Rebuttal · Authors · 2026-03-31
>
> Thank you for your detailed review. We believe that we have addressed each concern, and we hope that you will consider re-evaluating your score.
>
> ## Weaknesses
>
> 1. We have addressed your concerns in the bullet points below. Please let us know if there are extra points that we can resolve.
> 2. We appreciate your feedback and are curious if you could be a little more specific. We are not infallible, but we thought that we had done a reasonable job of this. We mentioned throughout the paper—abstract, introduction, background, and discussion—the assumptions of linearity and data-alignment. We also devoted a paragraph of the discussion to the data-aligned limitation. Please let us know if we can do more.
> 3. In our view, the whitened input limitation was not a severe one for the paper. The reason for this is because it is common to assume iid inputs when analyzing RNNs (Zucchet & Orvieto 2024; Hardt, Ma, and Recht 2018), and because the student still has to learn the non-trivial dynamics caused by the weights of the teacher transforming and adding temporal correlations to the white-noise. However, we thank the reviewer for bringing this up, and have added a section to the discussion mentioning the whitened-input limitation.
> 4. We thank the reviewer for this point and agree that our initial submission did not explain this in sufficient detail. Accordingly, we have completely revised section 2.2 to better align with the definition of data alignment in Proca et al. and to present our assumptions more clearly. In brief, it is assumed that the input-output correlation matrix of the data at time $t$ can be written as $\Sigma_t^{*}=US_tV^\top$, where $S_t$ is diagonal for all time steps (Proca et al.’s assumption). This assumption imposes relationships among the teacher parameters, given by the expressions on the right-hand side of our Eq. 11. Here, the teacher is data-aligned in the sense that its recurrent dynamics couple corresponding input–output modes without mixing across modes. For the student, it is assumed that the expressions on the left-hand side of Eq. 11 hold at initialization. The student is then aligned with the input-output modes of the teacher, and its recurrent dynamics likewise do not induce mode mixing. In section 2.3, our diagonalization of the expected learning updates shows that this alignment persists beyond initialization. The revised text conveys these points more directly.
> 5. We have fixed the brackets. At line 108 of the initial submission, right column, it is stated that expectations are taken over the input distribution; this holds throughout the paper.
> 6. A sum over $s$ is not missing; rather, the exponent of $\bar{W}$ should be $t$ instead of $t-s-1$. This is now fixed; thanks for the catch.
> 7. Fixed
> 8. We have not used boldface vectors as it is just as common in the literature to not use them—e.g. in linear algebra textbooks at various levels (e.g., Lang (1987) and Strang (1970)).
> 9. We have completely revised the Results section based on your comment. For each subsection, we made sure to first detail the methods, then present the results in the most straightforward way, and conclude with a clear takeaway. The most significant changes are in the subsection “Fixed-point manifolds”, which we rewrote completely. Please let us know if you would like us to elaborate.
> 10. Although we agree that the supplementary is dense, we do not believe it presents equations without sufficient guidance. As similar calculations are carried out for three learning rules, many steps reoccur. Accordingly, some derivations are presented once – for example, at the beginning of Appendix E – and then reused across learning rules. That said, certain sections would benefit from additional context at the cost of some repetition. We have thus added more explanatory details and clearer introductory remarks. We have also made the appendix more self-contained.
> 11. Now added: “Cyan lines and dots: optimal manifold; red lines and dots: non-optimal manifold.” to the caption.
> 12. Thank you, we have updated the text.
> 13. Done.
> ## Questions
> 1. As 1-step tBPTT struggles to perform as well as BPTT in most real-world problems, we believe that the data-aligned assumption is critical for this result. For this reason, we see the result as intriguing not because it provides a complete description of tBPTT and BPTT dynamics in general—indeed, it does not—but because it says that the difficult part of temporal learning is in learning to align with data, or learning temporal structure that does not satisfy our data-aligned assumptions—i.e., that requires the interaction of multiple computational modes. This is mentioned in the discussion.
> 2. Thank you for the references. We have incorporated them in the manuscript.
> 3. We stated this specifically with neuromorphic applications in mind, or other applications where local learning rules might be used to boost compute efficiency. We will make this context clear in the final sentence.

---

> > ### Author Rebuttal · Reviewer_zH6L · 2026-04-04
> >
> > I would like to thank the authors for responding to my and to the other reviewer's reviews.
> >
> > As I stated above, I find the topic interesting, and I would have loved to read this paper in a form that was ready for acceptance. However, in its current version, several of my main concerns remain only partially resolved, and the remaining issues are not ones that can be adequately addressed within a short rebuttal.
> >
> > **1]** Regarding the presentation: the results section and the derivations in the supplement rewrite is promised, but not yet available in the paper, and this was one of the central concerns in my review. I know ICML does not allow paper resubmission, but based on what was submitted I do not think the paper is ready to be published now.
> >
> > **2]** Although I understand that this is mainly a theory paper, the authors did not probe to see how far the theory works when violating the simplifying assumptions and directly test whether their conclusions survive beyond the considered setting, although they were asked both by me and by Reviewers XwWR and NdRD. For instance, they could explore controlled departure from data alignment, slightly non-normal recurrent matrices, nonlinear networks, finite-time experiments, or variance over feedback matrix (R) realisations (Fig. 5).
> >
> > **3]** Regarding the exaggeration of the findings. Some claims in the discussion read more broadly than the theory supports. The results of the paper are obtained under a restrictive data-aligned, symmetric-recurrent matrices with linear dynamics. Yet, the paper draws several broad conclusions about temporal credit assignment and learned representational structure, which are only suggestive rather than established outside the studied setting
> >  - **BPTT vs one-step tBPTT similarity**. What the paper shows is that, under the data-aligned assumption, the learning dynamics reduce to decoupled scalar mode equations, and in that regime BPTT and one-step tBPTT share the same fixed-point manifolds and have closely related local stability properties. But that result depends on a strong restriction, as the paper itself notes, of real symmetric recurrent matrices. Thus the analysis excludes generic non-normal recurrent dynamics, transient amplification, which are some of the phenomena that make RNN learning hard.
> > - **Temporal credit assignment.** In the paper, the distinction between BPTT, tBPTT, and RFLO is derived from how they weight lag-wise mismatch terms $E_t = AW^tB - A^\star (W^\star)^t B^\star$ and from the resulting scalar long-time ODEs. That is an ok analysis, but it is tied to the very specific setting of **linear student–teacher dynamics, terminal-time squared loss, iid Gaussian inputs, and long-T limits in a stable regime**. So, when the discussion suggests that the results imply sth general about how local rules handle temporal credit assignment, that should be read as a hypothesis suggested by the aligned linear theory, not as how recurrent learning unfolds under more realistic task structures, nonlinearities, or nonterminal losses. The discussion does note that more work is needed to test the broader interpretation, but some of the framing still reads as though the paper had established a more general principle than the current theory strictly supports.
> > - **Learned representation structure.** The paper has one strong theorem: for RFLO with $W_c=\hat w I$, the learned recurrent perturbation satisfies $W_K = W_0 + R^\top Q, \quad \mathrm{rank}(W_K-W_0)\le o$,
> > so the RFLO low-rank constraint holds even outside the aligned setting. But broader claims about representational structure are not established. In particular, the observation that one-step tBPTT also tends to learn low-rank perturbations is presented empirically in a very narrow synthetic setting (n=20, n*=10, scalar output, particular eigenvalue structure), and is not proven. So the paper can safely claim “RFLO provably imposes a low-rank inductive bias, and tBPTT appears to show a related tendency in our experiments,” but it should be more careful about implying a general theory of rank for approx. temp. credit assignment.
> > - **Fixed-point geometry.** (minor) Some of the discussion can leave the impression that 1-step tBPTT preserves the global learning geometry of exact gradient descent more generally than is shown.
> >
> > > Now added: “Cyan lines and dots: optimal manifold; red lines and dots: non-optimal manifold.” to the caption.
> >
> > It would be more helpful for understanding to add: optimal manifold (stable); non-optimal manifold (saddle) in the caption.
> >
> > **Overall, I would advise the authors to reflect more carefully on what they originally set out to achieve with this project and on why they chose to submit it in a premature form. In the next revision, they should articulate their findings and observations much more clearly, moderate the strength of their claims & include more experimental validation.**
> >
> > I will have to think more to decide whether I will update my recommendation.

---

> > > ### Author Response · Authors · 2026-04-08
> > >
> > > Thank you for the attention to detail in your reviewing of our paper. Please see our responses:
> > >
> > > 1] clarity of presentation is very important to us and we believe that, with the help of your comments, we were able to address these concerns. We agree that it is unfortunate that we cannot upload the new manuscript for you to see, but it is not uncommon in the reviewing process at ICML to see writing and presentation related changes made in this manner.
> > >
> > > 2] we agree that this is an exciting and important follow-up, but, in light of previous work like [Proca et al. ICML 2025](https://openreview.net/forum?id=KGOcrIWYnx) (see responses to reviewer XwWR for more details), we believe this is out of scope of our paper. We invite you to read our responses to the other reviewers addressing common concerns, and leading to adjusted scores.
> > >
> > > 3] We agree that exaggeration of results would be a serious problem, and we also believe that this is precisely the kind of fix that can be easily made during the rebuttal period. To avoid any exaggerated interpretation of our results we have changed our framing to address each of your points. Specifically:
> > >
> > > - bullet points 1 and 2: we are replacing discussion text from line 383 (left column) to the end of the same paragraph:
> > >
> > > *“The similar solutions of tBPTT and BPTT on *linear data-aligned tasks* suggest that we must move beyond the data-aligned regime to properly study temporal credit assignment. This is interesting because data-aligned RNNs can still exhibit long correlation time-scales and the “curse of memory” (cite: [Li et al JMLR 2022](https://www.jmlr.org/papers/v23/21-0368.html), [Zucchet & Orvieto NeurIPS 2024](https://proceedings.neurips.cc/paper_files/paper/2024/hash/fbb07254ef01868967dc891ea3fa6c13-Abstract-Conference.html)), highlighting that the curse of memory does not, in general, yield difficult temporal credit assignment problems. It also suggests that tasks that are difficult for temporal credit assignment are ones where multiple dynamic modes interact, rather than behaving independently with short or long timescales. Of course, we emphasize that more work is necessary to test this idea outside the linear, data-aligned regime.”*
> > >
> > > - bullet point 3: we change the fourth contribution in the introduction from *”Using numerical experiments, we also find empirically that this restriction affects tBPTT”* to *”Our numerical work also suggests that this restriction may affect tBPTT, at least in linear settings”*
> > >
> > > Lastly, to entirely resolve your concern about our results generalizing beyond the non-terminal loss, we added an extra supplementary section showing that all our results generalize to a sequence loss. The proof is below.
> > >
> > > Consider a sequence loss of the form:
> > >
> > > $\mathcal{L} = \frac{1}{2N}\sum_{T=1}^N||y_T - y_T^\ast||^2$.
> > >
> > > The sequence-loss update for each algorithm, for parameter $\theta$, has the following sample mean-type form:
> > >
> > > $$
> > > \begin{align}
> > >     \Delta_\mathcal{L}\theta = \frac{1}{N}\sum_{T=1}^N \Delta(\theta,T),
> > > \end{align}
> > > $$
> > >
> > > where $\Delta(\theta,T)$ is the update according to the loss $L_T$ for the given parameter and algorithm as in the main body of the paper.
> > >
> > > Our main results from sections 3.1-2 imply that an arbitrary element of the update array, $\Delta(\theta,T)$, converges to a value $\Delta\theta$ as $T\to\infty$. We can use this to show that in the large $N$ limit the sequence loss, $\mathcal{L}$, gives the same element-wise updates as the large-$T$ limit of the point-wise loss, $L_T$, used in the body of the paper:
> > >
> > > $$
> > > \lim_{N\to\infty}\Delta_\mathcal{L}\theta = \Delta\theta + \lim_{N\to\infty}\frac{1}{N}\sum_{T=1}^N[\Delta(\theta,T) - \Delta\theta],
> > > $$
> > >
> > > Let $\epsilon > 0$ be arbitrary. The last term converges to zero because, by the convergence of $\Delta(\theta,T)$, we can find and fix $S$ such that, for all $T \geq S$, $|\Delta(\theta,T) - \Delta\theta| < \frac{\epsilon}{2}$. Then
> > >
> > > \begin{align}
> > >     &\bigg|\frac{1}{N}\sum_{T=1}^N[\Delta(\theta,T) - \Delta\theta]\bigg| \leq \frac{1}{N}\sum_{T=1}^S\big|\Delta(\theta,T) - \Delta\theta\big| + \frac{1}{N}\sum_{T=S+1}^N\big|\Delta(\theta,T) - \Delta\theta\big| \leq \frac{1}{N}C_S + \frac{\epsilon}{2},
> > > \end{align}
> > >
> > > where $C_S$ does not depend on $N$. Selecting $N_0 \geq \frac{2C_S}{\epsilon}$ shows converges to zero, by definition, and completes the generalization of our results on data-aligned networks to the sequence loss.
> > >
> > > The low rank results of section 3.4 can also be easily shown to generalize to the sequence loss—the only difference being that an extra sum (over $T$) and a factor of $1/N$ appear in the $Q$ and $Q^{(b)}$ expressions.
> > >
> > > We believe that we have significantly addressed many of the concerns from your original review and rebuttal response. With these updates, and knowing that you share our view that our “paper tackles a rather interesting and exciting topic”, we wonder if you might consider updating your score to reflect these changes that you have helped us make.

---

### Official Review · Reviewer_XwWR · 2026-03-16

**Soundness:** 3
**Presentation:** 2
**Significance:** 2
**Originality:** 2
**Overall Recommendation:** 3
**Confidence:** 3

**Summary:**

This paper studies the infinite-data continuous-time limit of three algorithm for performing gradient descent on linear RNNs. Specifically, the authors consider backpropagation through time (BPTT), truncated BPPT (tBPTT), and random feedback local online (RFLO), with the goal of understanding the effects of local algorithms (tBPTT and RFLO) compared to the global one (BPTT). In a "data-aligned" teacher-student setup, the authors study properties such as fixed points, stability, convergence rate, and rank of the updates. They show that tBPTT mostly behaves similar to BPTT, while RFLO can be drastically different.

**Compliance With Llm Reviewing Policy:**

Affirmed.

**Final Justification:**

Overall, I still find the setting a bit too restrictive, and feel that there might not be enough new techniques to learn from the calculations, which is why I'm not recommending acceptance. But since I'm not familiar with this line of work and there might be an ICML audience interested in this work, I'll increase my initial score.

**Key Questions For Authors:**

My main concerns were stated above. Here are more minor questions/suggestion.

1. For readers like me who are unfamiliar with RFLO, it might be helpful to include an intuition about why this particular form of update is chosen.
2. Why is the loss measuring only the discrepancy at time $T$ and not throughout the process?
3. In Section 2.4, are $a,b,w$ eigenvalues of the solution to the ODE corresponding to $a_*,b_*,w_*$, at $T \to \infty$? This wasn't immediately clear and maybe it can be better presented.

**Limitations:**

yes

**Strengths And Weaknesses:**

### Strengths
The authors cover multiple aspects of the gradient descent dynamics induced by the three different algorithms, and are able to provide an extensive characterization of certain properties such as fixed points and stability. In cases where they can't characterize the behavior analytically, the authors provide numerical experiments to complete the picture.

### Weaknesses
I'm not an expert in this area, but to me it seems like the setting the paper chooses is a bit limited. It is not clear how much of the analysis and intuitions provided here carry over to more challenging settings where e.g. we no longer have aligned student and teacher. The analytical contributions seem to be calculations tailored to this specific structure rather than fundamental intuitions that could be more broadly useful, and on their own don't seem to be particularly strong for the ICML venue. The experiments are also limited to the same data-aligned setup.

I think there are two ways to improve the paper:
1. By considering a fundamentally more challenging theoretical setup and developing novel mathematical techniques there to come up with stronger formal statements.
2. By taking a more practical approach and having extensive experiments in more realistic setups.

---

> ### Author Rebuttal · Authors · 2026-03-31
>
> Thank you for your helpful feedback and questions. We believe that our manuscript has been improved by addressing your insights, and we hope that you might consider re-evaluating your score in light of these revisions. Please see below for our responses.
>
> ## Weaknesses
> 1. We agree that a mathematical framework for treating the non-data-aligned case would be exciting. However, we believe that our study is a worthy contribution as our assumptions are largely in line with – and in certain cases extend – those commonly used when studying learning dynamics in RNNs. For example:
> - [Schuessler et al. NeurIPS 2020](https://proceedings.neurips.cc/paper/2020/hash/9ac1382fd8fc4b631594aa135d16ad75-Abstract.html) study a linear RNN at equilibrium learning to produce a constant 1-dimensional output. Their theory contributes valuable information about the rank of solutions learned by gradient descent, and represents a step beyond this by considering RNNs learning constant outputs.
> - [Hardt et al. JMLR 2018](​​https://www.jmlr.org/papers/v19/16-465.html) prove convergence in single-input-single-output networks with the constraint that recurrent matrices are in controllable canonical form. Notably, this is also a strong constraint on the structure of the recurrent matrix
> - [Proca et al. ICML 2025](https://openreview.net/forum?id=KGOcrIWYnx) employ data-aligned assumptions to provide novel insight into RNN learning dynamics under BPTT in their ICML spotlight paper from last year. Our paper uses the same assumptions, only in a student teacher setting which does not significantly decrease generality and allows us to derive cleaner expressions in the long-input limit. Further, we believe that our work substantially builds on this by deriving results for RFLO and tBPTT, and characterizing fixed point stability.
> - [Zucchet & Orvieto NeurIPS 2024](https://proceedings.neurips.cc/paper_files/paper/2024/hash/fbb07254ef01868967dc891ea3fa6c13-Abstract-Conference.html) study loss function structure during learning in diagonal networks—thus networks that are, by definition, data-aligned.
> - [Li et al JMLR 2022](https://www.jmlr.org/papers/v23/21-0368.html) also study optimization of diagonal networks, this time learning linear functionals. Thus the structure of the learning models in this study are quite similar to our data-aligned student models
> Our work does not, of course, solve the learning problem in local neural networks in general, but the insights it provides should yield a useful stepping stone for future work that aims to relax these assumptions—just as the valuable above-cited works have done.
> 2. We agree with the reviewer about connecting our results to realistic settings, but we also believe there is value in studying simpler examples to build intuition, in the tradition of the references that we have added in the answer to the previous question.
>
> ## Questions
> 1. We have revised the first subsection of the Methods to clarify the three learning rules. Appendix B already describes the learning rules in detail, in particular the rationale for our formulation of RFLO. The remark at the end of section 2.1 also contains a high-level justification for our formulation. Appendix B is now mentioned more explicitly: “Appendix B provides additional details on the three learning rules.”
> 2. The initial goal in choosing this criterion was to simplify the derivations and to align with motor control and other neuroscience tasks, where performance is often evaluated at the end. This final-step loss is also the one used by Proca et al.
> 3. $a$, $b$, and $w$ are the parameters of the system under the change of basis that results in the diagonalized system described in section 2.3. That is, they correspond to the diagonal elements of the diagonal matrices that characterize the system, such that $a = \bar{A}_i$$_i$, $b = \bar{B}_i$$_i$, and $w = \bar{W}_i$$_i$, for some index $i$. Since each of these elements do not interact with each other in the RNN under the change of basis, we can study each eigenmode’s learning dynamics separately. And since the index $i$ is arbitrary we simply drop it, for ease of exposition, in the analysis. We have added a clearer explanation of this in the revised manuscript, in particular the following text in the Methods:
> “For each decoupled mode, we denote the student parameters by the triplet $(a, b, w)$ – arbitrary diagonal entries of $\bar{A}$,  $\bar{B}$, $\bar{W}$ – and the corresponding teacher parameters by $(a^\ast, b^\ast, w^\ast)$.” Please let us know if this helps!

---

> > ### Author Rebuttal · Reviewer_XwWR · 2026-04-03
> >
> > Thank you for your responses. I had a look at the references, and there seemed to be more technical intuition in the calculations compared to this work. Overall, I still find the setting a bit too restrictive, and feel that there might not be enough new techniques to learn from the calculations, which is why I'm not recommending acceptance. But since I'm not familiar with this line of work and there might be an ICML audience interested in this work, I'll increase my initial score.

---

> > > ### Author Response · Authors · 2026-04-08
> > >
> > > Thank you very much for the careful reviewing, and adjusting your score! We would be curious to hear a little more about two of your points: (1) that previous work contained more technical intuition; (2) that the setting might be too restrictive, even in light of previous work.
> > >
> > > To the first point, while we indeed make strong assumptions in our paper, we are able to use these assumptions to establish rigorous results—results which we believe are unique in the literature. Specifically, we have not seen other studies that have been able to perform fixed point stability analyses for learning dynamics in RNNs. With that in mind, would the reviewer be able to provide more insight into how our results do not provide sufficient technical intuition?
> > >
> > > To the second point, while we agree that different papers have used different analyses and this makes comparing restrictiveness somewhat more subjective, we would be curious to hear your thoughts on the comparison between our paper and [Proca et al. ICML 2025](https://openreview.net/forum?id=KGOcrIWYnx) in particular. This paper studied BPTT with the same linear data-aligned assumptions that we used, and was well received—it was accepted as a spotlight paper at ICML last year. This suggests to us that there is interest in the ICML community in using such strong assumptions to derive rigorous results, and we believe we have meaningfully built on this paper in three ways: (1) we analyzed two extra algorithms not present in the original paper (tBPTT and RFLO); (2) we derived clean expressions for learning, based on series expansions, which eliminate the summations used in the original paper; (3) we used dynamical systems theory to perform stability analyses for all three algorithms in a way that seems, to us, to be novel in the RNN optimization literature.
> > >
> > > We also wish to emphasize that our results in section 3.4 on the rank of solutions learned by RFLO are entirely free of the data aligned assumptions, and thus apply quite generally to learning in linear RNNs.
> > >
> > > Finally, we are excited to highlight that, on the request of another reviewer, we have generalized all our results from a single-time point loss to a sequence loss (see response to reviewer zH6L).
> > >
> > > We appreciate your uncertainty, with limited familiarity in this line of work (we can deeply relate to this given our experience reviewing at the big three conferences!), and we wonder if this comparison with past work—along with two other reviewers now advocating for acceptance—might help convince you that our paper could be of interest to the community, and thus might lead you to reconsider your score. Thank you again for the attentive reviewing of our work.

---

### Decision · Program_Chairs · 2026-04-30

**Decision:**

Accept (regular)

**Comment:**

The reviews are mixed. The main recurring complaint is the fact that the setting is quite restricted (linear, diagonal/aligned). A lot of other concerns (about clarity) have been addressed in the discussion. In spite of the simplicity of the setting, the paper provides some useful insight into the effect of locality on RNN training.